# DISCRIMINATIVE REPRESENTATION LOSS (DRL): A MORE EFFICIENT APPROACH THAN GRADIENT RE-PROJECTION IN CONTINUAL LEARNING

## ABSTRACT

The use of episodic memories in continual learning has been shown to be effective in terms of alleviating catastrophic forgetting. In recent studies, several gradient-based approaches have been developed to make more efficient use of compact episodic memories, which constrain the gradients resulting from new samples with those from memorized samples, aiming to reduce the diversity of gradients from different tasks. In this paper, we reveal the relation between diversity of gradients and discriminativeness of representations, demonstrating connections between Deep Metric Learning and continual learning. Based on these findings, we propose a simple yet highly efficient method – Discriminative Representation Loss (DRL) – for continual learning. In comparison with several state-of-the-art methods, DRL shows effectiveness with low computational cost on multiple benchmark experiments in the setting of online continual learning.

## 1 INTRODUCTION

In the real world, we are often faced with situations where data distributions are changing over time, and we would like to update our models by new data in time, with bounded growth in system size. These situations fall under the umbrella of "continual learning", which has many practical applications, such as recommender systems, retail supply chain optimization, and robotics (Lesort et al., 2019; Diethe et al., 2018; Tian et al., 2018). Comparisons have also been made with the way that humans are able to learn new tasks without forgetting previously learned ones, using common knowledge shared across different skills. The fundamental problem in continual learning is *catastrophic forgetting* (McCloskey & Cohen, 1989; Kirkpatrick et al., 2017), *i.e.* (neural network) models have a tendency to forget previously learned tasks while learning new ones.

There are three main categories of methods for alleviating forgetting in continual learning: *i*) regularization-based methods which aim in preserving knowledge of models of previous tasks (Kirkpatrick et al., 2017; Zenke et al., 2017; Nguyen et al., 2018) *ii*) architecture-based methods for incrementally evolving the model by learning task-shared and task-specific components (Schwarz et al., 2018; Hung et al., 2019); *iii*) replay-based methods which focus in preserving knowledge of data distributions of previous tasks, including methods of experience replay by episodic memories or generative models (Shin et al., 2017; Rolnick et al., 2019), methods for generating compact episodic memories (Chen et al., 2018; Aljundi et al., 2019), and methods for more efficiently using episodic memories (Lopez-Paz & Ranzato, 2017; Chaudhry et al., 2019a; Riemer et al., 2019; Farajtabar et al., 2020).

Gradient-based approaches using episodic memories, in particular, have been receiving increasing attention. The essential idea is to use gradients produced by samples from episodic memories to constrain the gradients produced by new samples, *e.g.* by ensuring the inner product of the pair of gradients is non-negative (Lopez-Paz & Ranzato, 2017) as follows:

$$\langle g_t, g_k \rangle = \left\langle \frac{\partial \mathcal{L}(x_t, \theta)}{\partial \theta}, \frac{\partial \mathcal{L}(x_k, \theta)}{\partial \theta} \right\rangle \geq 0, \ \ \forall k < t \tag{1}$$

where $t$ and $k$ are time indices, $x_t$ denotes a new sample from the current task, and $x_k$ denotes a sample from the episodic memory. Thus, the updates of parameters are forced to preserve the performance on previous tasks as much as possible.

In Gradient Episodic Memory (GEM) (Lopez-Paz & Ranzato, 2017), $g_t$ is projected to a direction that is closest to it in $L_2$-norm whilst also satisfying Eq. (1): $\min_{\tilde{g}} \frac{1}{2}||g_t - \tilde{g}||_2^2, \quad s.t. \langle \tilde{g}, g_k \rangle \geq 0, \quad \forall k < t$. Optimization of this objective requires a high-dimensional quadratic program and thus is computationally expensive. Averaged-GEM (A-GEM) (Chaudhry et al., 2019a) alleviates the computational burden of GEM by using the averaged gradient over a batch of samples instead of individual gradients of samples in the episodic memory. This not only simplifies the computation, but also obtains comparable performance with GEM. Orthogonal Gradient Descent (OGD) (Farajtabar et al., 2020) projects $g_t$ to the direction that is perpendicular to the surface formed by $\{g_k | k < t\}$. Moreover, Aljundi et al. (2019) propose Gradient-based Sample Selection (GSS), which selects samples that produce most diverse gradients with other samples into episodic memories. Here diversity is measured by the cosine similarity between gradients. Since the cosine similarity is computed using the inner product of two normalized gradients, GSS embodies the same principle as other gradient-based approaches with episodic memories. Although GSS suggests the samples with most diverse gradients are important for generalization across tasks, Chaudhry et al. (2019b) show that the average gradient over a small set of random samples may be able to obtain good generalization as well.

In this paper, we answer the following questions: *i*) Which samples tend to produce diverse gradients that strongly conflict with other samples and why are such samples able to help with generalization? *ii*) Why does a small set of randomly chosen samples also help with generalization? *iii*) Can we reduce the diversity of gradients in a more efficient way? Our answers reveal the relation between diversity of gradients and discriminativeness of representations, and further show connections between Deep Metric Learning (DML) (Kaya & Bilge, 2019; Roth et al., 2020) and continual learning. Drawing on these findings we propose a new approach, Discriminative Representation Loss (DRL), for classification tasks in continual learning. Our methods show improved performance with relatively low computational cost in terms of time and RAM cost when compared to several state-of-the-art (SOTA) methods across multiple benchmark tasks in the setting of online continual learning.

## 2 A New Perspective of Reducing Diversity of Gradients

According to Eq. (1), negative cosine similarities between gradients produced by current and previous tasks result in worse performance in continual learning. This can be interpreted from the perspective of constrained optimization as discussed by Aljundi et al. (2019). Moreover, the diversity of gradients relates to the Gradient Signal to Noise Ratio (GSNR) (Liu et al., 2020), which plays a crucial role in the model's generalization ability. Intuitively, when more of the gradients point in diverse directions, the variance will be larger, leading to a smaller GSNR, which indicates that reducing the diversity of gradients can improve generalization. This finding leads to the conclusion that samples with the most diverse gradients contain the most critical information for generalization which is consistent with in Aljundi et al. (2019).

### 2.1 The source of gradient diversity

We first conducted a simple experiment on classification tasks of 2-D Gaussian distributions, and tried to identify samples with most diverse gradients in the 2-D feature space. We trained a linear model on the first task to discriminate between two classes (blue and orange dots in Fig. 1a). We then applied the algorithm Gradient-based Sample Selection with Interger Quadratic Programming (GSS-IQP) (Aljundi et al., 2019) to select 10% of the samples of training data that produce gradients with the lowest similarity (black dots in Fig. 1a), and denote this set of samples as $\widehat{M} = \min_M \sum_{i,j \in M} \frac{\langle g_i, g_j \rangle}{||g_i|| \cdot ||g_j||}$.

It is clear from Fig. 1a that the samples in $\widehat{M}$ are mostly around the decision boundary between the two classes. Increasing the size of $\widehat{M}$ results in the inclusion of samples that trace the outer edges of the data distributions from each class. *Clearly the gradients can be strongly opposed when samples from different classes are very similar*. Samples close to decision boundaries are most likely to exhibit this characteristic. Intuitively, storing the decision boundaries of previously learned classes should be an effective way to preserve classification performance on those classes. However, if the episodic memory only includes samples representing the learned boundaries, it may miss important information when the model is required to incrementally learn new classes. We show this by introducing a second task - training the model above on a third class (green dots). We display the decision boundaries (which split the feature space in a one *vs.* all manner) learned by the model after

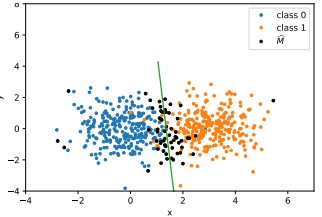 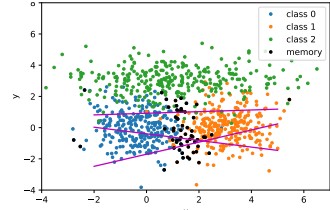 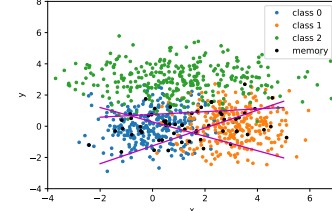

(a) Samples with most diverse gradients ($\widehat{M}$) after learning task 1, the green line is the decision boundary.

(b) Learned decision boundaries (purple lines) after task 2. Here the episodic memory includes samples in $\widehat{M}$.

(c) Learned decision boundaries (purple lines) after task 2. Here the episodic memory consists of random samples.

Figure 1: 2-D classification examples, the $x$ and $y$ axis are the coordinates (also features) of samples. We sequentially train a logistic regression model on two tasks: the first task is to classify two classes as shown in (a); the second class is to incrementally classify a third class as shown in (b) and (c). The solid lines are decision boundaries between classes.

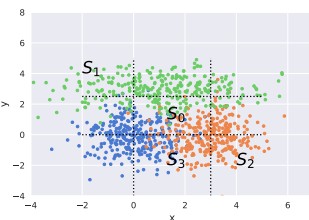 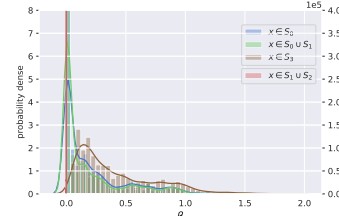 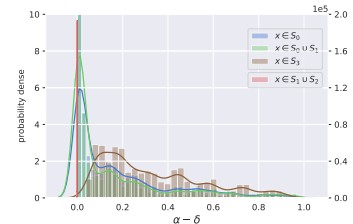

(a) Splitting samples into several subsets in a 3-class classification task. Dots in different colors are from different classes.

(b) Estimated distributions of $\beta$ when drawing negative pairs from different subsets of samples.

(c) Estimated distributions of $\alpha - \delta$ when drawing negative pairs from different subsets of samples.

Figure 2: Illustration of how $\Pr(2\beta > \alpha - \delta)$ in Theorem 1 behaves in various cases by drawing negative pairs from different subsets of a 3-class feature space which are defined in Fig. 2a. The classifier is a linear model. y-axis in the right side of (b) & (c) is for the case of $x \in S_1 \cup S_2$. We see that $\alpha - \delta$ behaves in a similar way with $\beta$ but in a smaller range which makes $\beta$ the key in studying $\Pr(2\beta > \alpha - \delta)$. In the case of $x \in S_3$ the distribution of $\beta$ has more mass on larger values than other cases because the predicted probabilities are mostly on the two classes in a pair, and it causes all $\langle \mathbf{g}_n, \mathbf{g}_m \rangle$ having the opposite sign of $\langle \mathbf{x}_n, \mathbf{x}_m \rangle$ as shown in Tab. 1.

task 2 with $\widehat{M}$ (Fig. 1b) and a random set of samples (Fig. 1c) from task 1 as the episodic memory. The random episodic memory shows better performance than the one selected by GSS-IQP, since the new decision boundaries rely on samples not included in $\widehat{M}$. It explains why randomly selected memories may generalize better in continual learning. Ideally, with $\widehat{M}$ large enough, the model can remember all edges of each class, and hence learn much more accurate decision boundaries sequentially. However, memory size is often limited in practice, especially for high-dimensional data. A more efficient way could be learning more informative representations. The experimental results indicate that: *1) more similar representations in different classes result in more diverse gradients. 2) more diverse representations in a same class help with learning new boundaries incrementally.*

Now we formalise the connection between the diversity of gradients and the discriminativeness of representations for the linear model (proofs are in Appx. A). **Notations**: *Negative pair* represents two samples from different classes. *Positive pair* represents two samples from a same class. Let $\mathcal{L}$ represent the softmax cross entropy loss, $\mathbf{W} \in \mathbb{R}^{D \times K}$ is the weight matrix of the linear model, and $\mathbf{x}_n \in \mathbb{R}^D$ denotes the input data, $\mathbf{y}_n \in \mathbb{R}^K$ is a one-hot vector that denotes the label of $\mathbf{x}_n$, $D$ is the dimension of representations, $K$ is the number of classes. Let $\boldsymbol{p}_n = softmax(\mathbf{o}_n)$, where $\mathbf{o}_n = \mathbf{W}^T \mathbf{x}_n$, the gradient $\boldsymbol{g}_n = \nabla_{\mathbf{W}} \mathcal{L}(\mathbf{x}_n, \mathbf{y}_n; \mathbf{W})$. $\mathbf{x}_n, \mathbf{x}_m$ are two different samples when $n \neq m$.

**Lemma 1.** *Let $\boldsymbol{\epsilon}_n = \boldsymbol{p}_n - \mathbf{y}_n$, we have:* $\langle \boldsymbol{g}_n, \boldsymbol{g}_m \rangle = \langle \mathbf{x}_n, \mathbf{x}_m \rangle \langle \boldsymbol{\epsilon}_n, \boldsymbol{\epsilon}_m \rangle$,

Table 1: Illustration of the Theorems by drawing pairs from different subsets that are defined in Fig. 2a. We obtain the gradients and predictions by a linear model and a MLP with two hidden layers (16 units for each) and ReLU (or tanh) activations. The gradients are computed by all parameters of the model. We can see that the non-linear models exhibit similar behaviors with the linear model as described in the theorems. One exception is that the MLP with ReLU activations gets much less negative $\langle g_n, g_m \rangle$ in the case of $S_1 \cup S_2$ for negative pairs, we consider the difference is caused by representations to the final linear layer always being positive in this case due to ReLU activations.

| | | Negative pairs (Thm. 1) | | | | Positive pairs (Thm.2) | | | |
|---|---|---|---|---|---|---|---|---|---|
| | | $S_0$ | $S_0 \cup S_1$ | $S_3$ | $S_1 \cup S_2$ | $S_0$ | $S_0 \cup S_1$ | $S_3$ | $S_1 \cup S_2$ |
| | $\Pr(\langle \mathbf{x}_n, \mathbf{x}_m \rangle > 0)$ | 1. | 0.877 | 1. | 0. | 1. | 0.99 | 1. | 1. |
| Linear | $\Pr(\langle g_n, g_m \rangle < 0)$ | 0.727 | 0.725 | 1. | 0.978 | 0. | 0.007 | 0. | 0. |
| | $\Pr(2\beta + \delta > \alpha)$ | 0.727 | 0.687 | 1. | 0. | − | − | − | − |
| MLP (ReLU) | $\Pr(\langle g_n, g_m \rangle < 0)$ | 0.72 | 0.699 | 1. | 0.21 | 0.013 | 0.01 | 0. | 0. |
| | $\Pr(2\beta + \delta > \alpha)$ | 0.746 | 0.701 | 1. | 0. | − | − | − | − |
| MLP (tanh) | $\Pr(\langle g_n, g_m \rangle < 0)$ | 0.745 | 0.744 | 1. | 0.993 | 0.004 | 0.007 | 0. | 0. |
| | $\Pr(2\beta + \delta > \alpha)$ | 0.766 | 0.734 | 1. | 0. | − | − | − | − |

**Theorem 1.** *Suppose* $\mathbf{y}_n \neq \mathbf{y}_m$, *and let* $c_n$ *denote the class index of* $\mathbf{x}_n$ *(i.e.* $\mathbf{y}_{n,c_n} = 1, \mathbf{y}_{n,i} = 0, \forall i \neq c_n$). *Let* $\alpha \triangleq ||p_n||^2 + ||p_m||^2$, $\beta \triangleq p_{n,c_m} + p_{m,c_n}$ *and* $\delta \triangleq ||p_n - p_m||_2^2$, *then:*

$$\Pr\left(sign(\langle g_n, g_m \rangle) = sign(-\langle \mathbf{x}_n, \mathbf{x}_m \rangle)\right) = \Pr(2\beta > \alpha - \delta),$$

**Theorem 2.** *Suppose* $\mathbf{y}_n = \mathbf{y}_m$, *when* $\langle g_n, g_m \rangle \neq 0$, *we have:* $sign(\langle g_n, g_m \rangle) = sign(\langle \mathbf{x}_n, \mathbf{x}_m \rangle)$

For a better understanding of the theorems, we conduct empirical study by partitioning the feature space of three classes into several subsets as shown in Fig. 2a and examine four cases of pairwise samples by these subsets: 1). $\mathbf{x} \in S_0$, both samples in a pair are near the intersection of the three classes; 2). $\mathbf{x} \in S_0 \cup S_1$, one sample is close to decision boundaries and the other is far away from the boundaries; 3). $\mathbf{x} \in S_3$, both samples close to the decision boundary between their true classes but away from the third class; 4). $\mathbf{x} \in S_1 \cup S_2$, both samples are far away from the decision boundaries.

Theorem 1 says that for samples from different classes, $\langle g_n, g_m \rangle$ gets an opposite sign of $\langle \mathbf{x}_n, \mathbf{x}_m \rangle$ with a probability that depends on the predictions $p_n$ and $p_m$. This probability of flipping the sign especially depends on $\beta$ which reflects how likely to misclassify both samples to its opposite class. We show the empirical distributions of $\beta$ and $(\alpha - \delta)$ obtained by a linear model in Figs. 2b and 2c, respectively. In general, $(\alpha - \delta)$ shows similar behaviors with $\beta$ in the four cases but in a smaller range, which makes $2\beta > (\alpha - \delta)$ tends to be true except when $\beta$ is around zero. Basically, a subset including more samples close to decision boundaries leads to more probability mass on large values of $\beta$, and the case of $\mathbf{x} \in S_3$ results in largest mass on large values of $\beta$ because the predicted probabilities mostly concentrate on the two classes in a pair. As shown in Tab. 1, more mass on large values of $\beta$ leads to larger probabilities of flipping the sign. These results demonstrate that samples with most diverse gradients (which gradients have largely negative similarities with other samples) are close to decision boundaries because they tend to have large $\beta$ and $\langle \mathbf{x}_n, \mathbf{x}_m \rangle$ tend to be positive. In the case of $\mathbf{x} \in S_1 \cup S_2$ the probability of flipping the sign is zero because $\beta$ concentrates around zero. According to Lemma 1 $\langle g_n, g_m \rangle$ are very close to zero in this case because the predictions are close to true labels, hence, such samples are not considered as with most diverse gradients.

Theorem 2 says $\langle g_n, g_m \rangle$ has the same sign as $\langle \mathbf{x}_n, \mathbf{x}_m \rangle$ when the two samples from a same class. We can see the results of positive pairs in Tab. 1 matches Theorem 2. In the case of $S_0 \cup S_1$ the two probabilities do not add up to exactly 1 because the implementation of cross-entropy loss in tensorflow smooths the function by a small value for preventing numerical issues which slightly changes the gradients. As $\langle \mathbf{x}_n, \mathbf{x}_m \rangle$ is mostly positive for positive pairs, $\langle g_n, g_m \rangle$ hence is also mostly positive, which explains why samples with most diverse gradients are not sufficient to preserve information within classes in experiments of Fig. 1. On the other hand, if $\langle \mathbf{x}_n, \mathbf{x}_m \rangle$ is negative then $\langle g_n, g_m \rangle$ will be negative, which indicates representations within a class should not be too diverse.

Extending this theoretical analysis based on a linear model, we also provide empirical study of non-linear models (Multi-layer Perceptrons (MLPs)). As demonstrated in Tab. 1, the probability of flipping the sign in MLPs are very similar with the linear model since it only depends on the predictions and all models have learned reasonable decision boundaries. The probability of getting

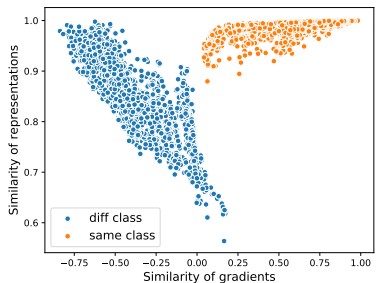 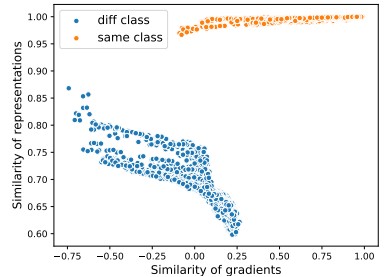

(a) Similarities of gradients *vs.* representations (class 7 & 9)

(b) Similarities of gradients *vs.* representations (class 0 & 1)

Figure 3: Similarities of gradients and representations of two classes in the MNIST dataset. The $x$ and $y$ axis are the cosine similarity of gradients and representations, respectively. Blue dots indicate the similarity of *negative pairs*, while orange dots indicate that of *positive pairs*.

negative $\langle \boldsymbol{g}_n, \boldsymbol{g}_m \rangle$ is also similar with the linear model except in the case of $S_1 \cup S_2$ for negative pairs, in which the MLP with ReLU gets much less negative $\langle \boldsymbol{g}_n, \boldsymbol{g}_m \rangle$. As MLP with tanh activations is still consistent with the linear model in this case, we consider the difference is caused by the representations always being positive due to ReLU activations. These results demonstrate that non-linear models exhibit similar behaviors with linear models that mostly align with the theorems.

Since only negative $\langle \boldsymbol{g}_n, \boldsymbol{g}_m \rangle$ may cause conflicts, reducing the diversity of gradients hence relies on reducing negative $\langle \boldsymbol{g}_n, \boldsymbol{g}_m \rangle$. We consider to reduce negative $\langle \boldsymbol{g}_n, \boldsymbol{g}_m \rangle$ by two ways: 1).minimize the representation inner product of negative pairs, which pushes the inner product to be negative or zero (for positive representations); 2).optimize the predictions to decrease the probability of flipping the sign. In this sense, decreasing the representation similarity of negative pairs might help with both ways. In addition, according to Fig. 2b $\mathbf{x} \sim S_3$ gets larger prediction similarity than $\mathbf{x} \sim S_0$ due to the predictions put most probability mass on both classes of a pair, which indicates decreasing the similarity of predictions may decrease the probability of flipping the sign. Hence, we include logits in the representations. We verify this idea by training two binary classifiers for two groups of MNIST classes ($\{0, 1\}$ and $\{7, 9\}$). The classifiers have two hidden layers each with 100 hidden units and ReLU activations. We randomly chose 100 test samples from each group to compute the pairwise cosine similarities. Representations are obtained by concatenating the output of all layers (including logits) of the neural network, gradients are computed by all parameters of the model. We display the similarities in Figs. 3a and 3b. The correlation coefficients between the gradient and representation similarities of negative pairs are -0.86 and -0.85, which of positive pairs are 0.71 and 0.79. In all cases, the similarities of representations show strong correlations with the similarities of gradients. The classifier for class 0 and 1 gets smaller representation similarities and much less negative gradient similarities for negative pairs (blue dots) and it also gains a higher accuracy than the other classifier (99.95% vs. 96.25%), which illustrates the potential of reducing the gradient diversity by decreasing the representation similarity of negative pairs.

## 2.2 CONNECTING DEEP METRIC LEARNING TO CONTINUAL LEARNING

Reducing the representation similarity between classes shares the same concept as learning larger margins which has been an active research area for a few decades. For example, Kernel Fisher Discriminant analysis (KFD) (Mika et al., 1999) and distance metric learning (Weinberger et al., 2006) aim to learn kernels that can obtain larger margins in an implicit representation space, whereas Deep Metric Learning (DML) (Kaya & Bilge, 2019; Roth et al., 2020) leverages deep neural networks to learn embeddings that maximize margins in an explicit representation space. In this sense, DML has the potential to help with reducing the diversity of gradients in continual learning.

However, the usual concepts in DML may not entirely be appropriate for continual learning, as they also aim in learning compact representations within classes (Schroff et al., 2015; Wang et al., 2017; Deng et al., 2019). In continual learning, the unused information for the current task might be important for a future task, e.g. in the experiments of Fig. 1 the y-dimension is not useful for task 1 but useful for task 2. It indicates that learning compact representations in a current task might omit important dimensions in the representation space for a future task. In this case, even if we

Table 2: Demonstration of performance degrading in continual learning by compact representations. We test the first two tasks of split-MNIST and split-Fashion MNIST by training a MLP (2 hidden layers with 100 units per layer and ReLU actiavtions) with and without L1 regularization. The memory is formed by 50 samples that are randomly chosen. Representations are outputs of hidden layers. We identify important dimensions of the representation space by selecting the hidden units that have a mean activation larger than 1 over all learned classes.

| | L1 | Avg. Accuracy (t=1) | Avg. Accuracy (t=2) | #. Important dimensions |
|---|---|---|---|---|
| S-MNIST | | 99.9 | 94.5 | 39 |
| | ✓ | 99.7 | 81.1 | 6 |
| S-Fashion | | 97.6 | 88.2 | 55 |
| | ✓ | 96.4 | 76.2 | 10 |

store diverse samples into the memory, the learned representations may be difficult to generalize on future tasks as the omitted dimensions can only be relearned by using limited samples in the memory. We demonstrate this by training a model with and without L1 regulariztion on the first two tasks of split-MNIST and split-Fashion MNIST. The results are shown in Tab. 2. We see that with L1 regularization the model learns much more compact representations and gives a similar performance on task 1 but much worse performance on task 2 comparing to without L1 regularization. The results suggest that continual learning shares the interests of maximizing margins in DML but prefers less compact representation space to preserve necessary information for future tasks. We suggest an opposite way regarding the within-class compactness: minimizing the similarities within the same class for obtaining less compact representation space. Roth et al. (2020) proposed a $\rho$-spectrum metric to measure the information entropy contained in the representation space (details are provided in Appx. D) and introduced a $\rho$-regularization method to restrain over-compression of representations. The $\rho$-regularization method randomly replaces negative pairs by positive pairs with a pre-selected probability $p_\rho$. Nevertheless, switching pairs is inefficient and may be detrimental to the performance in an online setting because some negative pairs may never be learned in this way. Thus, we propose a different way to restrain the compression of representations which will be introduced in the following.

## 3 DISCRIMINATIVE REPRESENTATION LOSS

Based on our findings in the above section, we propose an auxiliary objective Discriminative Representation Loss (DRL) for classification tasks in continual learning, which is straightforward, robust, and efficient. Instead of explicitly re-projecting gradients during training process, DRL helps with decreasing gradient diversity by optimizing the representations. As defined in Eq. (2), DRL consists of two parts: one is for minimizing the similarities of representations from different classes ($\mathcal{L}_{bt}$) which can reduce the diversity of gradients from different classes, the other is for minimizing the similarities of representations from a same class ($\mathcal{L}_{wi}$) which helps preserve discriminative information for future tasks in continual learning.

$$\min_{\Theta} \mathcal{L}_{DRL} = \min_{\Theta}(\mathcal{L}_{bt} + \alpha \mathcal{L}_{wi}), \ \ \alpha > 0,$$

$$\mathcal{L}_{bt} = \frac{1}{N_{bt}} \sum_{i=1}^{B} \sum_{j \neq i, y_j \neq y_i}^{B} \langle h_i, h_j \rangle, \quad \mathcal{L}_{wi} = \frac{1}{N_{wi}} \sum_{i=1}^{B} \sum_{j \neq i, y_j = y_i}^{B} \langle h_i, h_j \rangle, \tag{2}$$

where $\Theta$ denotes the parameters of the model, $B$ is training batch size. $N_{bt}, N_{wi}$ are the number of negative and positive pairs, respectively. $\alpha$ is a hyperparameter controlling the strength of $\mathcal{L}_{wi}$, $h_i$ is the representation of $x_i$, $y_i$ is the label of $x_i$. The final loss function combines the commonly used softmax cross entropy loss for classification tasks ($\mathcal{L}$) with DRL ($\mathcal{L}_{DRL}$) as shown in Eq. (3),

$$\widehat{\mathcal{L}} = \mathcal{L} + \lambda \mathcal{L}_{DRL}, \ \ \lambda > 0, \tag{3}$$

where $\lambda$ is a hyperparameter controlling the strength of $\mathcal{L}_{DRL}$, which is larger for increased resistance to forgetting, and smaller for greater elasticity. We provide experimental results to verify the effects of DRL and an ablation study on $\mathcal{L}_{bt}$ and $\mathcal{L}_{wi}$ (Tab. 7) in Appx. E, according to which $\mathcal{L}_{bt}$ and $\mathcal{L}_{wi}$

have shown effectiveness on improving forgetting and $\rho$-spectrum, respectively. We will show the correlation between $\rho$-spectrum and the model performance in Sec. 5.

The computational complexity of DRL is $O(B^2 H)$, where $B$ is training batch size, $H$ is the dimension of representations. $B$ is small (10 or 20 in our experiments) and commonly $H \ll W$, where $W$ is the number of network parameters. In comparison, the computational complexity of A-GEM and GSS-greedy are $O(B_r W)$ and $O(B B_m W)$, respectively, where $B_r$ is the reference batch size in A-GEM and $B_m$ is the memory batch size in GSS-greedy. The computational complexity discussed here is additional to the cost of common backpropagation. We compare the training time of all methods on MNIST tasks in Tab. 9 in Appx. H, which shows the representation-based methods require much lower computational cost than gradient-based approaches.

## 4 ONLINE MEMORY UPDATE AND BALANCED EXPERIENCE REPLAY

We follow the *online setting* of continual learning as was done for other gradient-based approaches with episodic memories (Lopez-Paz & Ranzato, 2017; Chaudhry et al., 2019a; Aljundi et al., 2019), in which the model only trained with one epoch on the training data.

We update the episodic memories by the basic ring buffer strategy: keep the last $n_c$ samples of class $c$ in the memory buffer, where $n_c$ is the memory size of a seen class $c$. We have deployed the episodic memories with a fixed size, implying a fixed budget for the memory cost. Further, we maintain a uniform distribution over all seen classes in the memory. The buffer may not be evenly allocated to each class before enough samples are acquired for newly arriving classes. We show pseudo-code of the memory update strategy in Alg. 1 in Appx. B for a clearer explanation. For class-incremental learning, this strategy can work without knowing task boundaries. Since DRL and methods of DML depend on the pairwise similarities of samples, we would prefer the training batch to include as wide a variety of different classes as possible to obtain sufficient discriminative information. Hence, we adjust the Experience Replay (ER) strategy (Chaudhry et al., 2019b) for the needs of such methods. The idea is to uniformly sample from seen classes in the memory buffer to form a training batch, so that this batch can contain as many seen classes as possible. Moreover, we ensure the training batch includes at least one positive pair of each selected class (minimum 2 samples in each class) to enable the parts computed by positive pairs in the loss. In addition, we also ensure the training batch includes at least one class from the current task. We call this Balanced Experience Replay (BER). The pseudo code is in Alg. 2 of Appx. B. Note that we update the memory and form the training batch based on the task ID instead of class ID for instance-incremental tasks (e.g. permuted MNIST tasks), as in this case each task always includes the same set of classes.

## 5 EXPERIMENTS

In this section we evaluate our methods on multiple benchmark tasks by comparing with several baseline methods in the setting of online continual learning.

**Benchmark tasks:** We have conducted experiments on the following benchmark tasks: *Permuted MNIST* (10 tasks and each task includes the same 10 classes with different permutation of features), *Split MNIST*, *Split Fashion-MNIST*, and *Split CIFAR-10* (all three having 5 tasks with two classes in each task), *Split CIFAR-100* (10 tasks with 10 classes in each task), *Split TinyImageNet* (20 tasks with 10 classes in each task). All split tasks include disjoint classes. For tasks of MNIST (LeCun et al., 2010) and Fashion-MNIST (Xiao et al., 2017), the training size is 1000 samples per task, for CIFAR-10 (Krizhevsky et al., 2009) the training size is 3000 per task, for CIFAR-100 and TinyImageNet (Le & Yang, 2015) it is 5000 per task. ***N.B.***: We use *single-head* (shared output) models in all of our experiments, meaning that we do not require a task identifier at testing time. Such settings are more difficult for continual learning but more practical in real applications.

**Baselines:** We compare our methods with: two gradient-based approaches (*A-GEM* (Chaudhry et al., 2019a) and *GSS-greedy* (Aljundi et al., 2019)), two standalone experience replay methods (*ER* (Chaudhry et al., 2019b) and *BER*), two SOTA methods of DML (*Multisimilarity* (Wang et al., 2019) and *R-Margin* (Roth et al., 2020)). We also trained a *single* task over all classes with **one epoch** for all benchmarks which performance can be viewed as a upper bound of each benchmark. ***N.B.***: We deploy the losses of Multisimilarity and R-Margin as auxiliary objectives as the same as DRL

because using standalone such losses causes difficulties of convergence in our experimental settings. We provide the definitions of these two losses in Appx. D.

**Performance measures:** We use the *Average accuracy*, *Average forgetting*, *Average intransigence* to evaluate the performance of all methods, the definition of these measures are provided in Appx. C

**Experimental settings:** We use the vanilla SGD optimizer for all experiments without any scheduling. For tasks on MNIST and Fashion-MNIST, we use a MLP with two hidden layers and ReLU activations, and each layer has 100 hidden units. For tasks on CIFAR datasets and TinyImageNet, we use the same reduced Resnet18 as used in Chaudhry et al. (2019a). All networks are trained from scratch without regularization scheme. For the MLP, representations are the concatenation of outputs of all layers including logits; for reduced Resnet18, representations are the concatenation of the input of the final linear layer and output logits. We concatenate outputs of all layers as we consider they behave like different levels of representation, and when higher layers (layers closer to the input) generate more discriminative representations it would be easier for lower layers to learn more discriminative representations as well. This method also improves the performance of MLPs. For reduced ResNet18 we found that including outputs of all hidden layers performs almost the same as only including the final representations, so we just use the final layer for lower computational cost. We deploy BER as the replay strategy for DRL, Multisimilarity, and R-Margin. The memory size for tasks on MNIST and Fashion-MNIST is 300 samples. For tasks on CIFAR-10 and CIFAR-100 the memory size is 2000 and 5000 samples, respectively. For TinyImageNet it is also 5000 samples. The standard deviation shown in all results are evaluated over 10 runs with different random seeds. We use 10% of training set as validation set for choosing hyperparameters by cross validation. More details of experimental settings and hyperparameters are given in Appx. I.

Tabs. 3 to 5 give the averaged accuracy, forgetting, and intransigence of all methods on all benchmark tasks, respectively. As we can see, the forgetting and intransigence often conflict with each other which is the most common phenomenon in continual learning. Our method DRL is able to get a better trade-off between them and thus outperforms other methods over most benchmark tasks in terms of average accuracy. This could be because DRL facilitates getting a good intransigence and $\rho$-spectrum by $\mathcal{L}_{wi}$ and a good forgetting by $\mathcal{L}_{bt}$. In DRL the two terms are complementary to each other and combining them brings benefits on both sides (an ablation study on the two terms are provide in Appx. E). According to Tabs. 4 and 5, Multisimilarity got better avg. intransigence and similar avg. forgetting on CIFAR-10 compared with DRL which indicates Multisimilarity learns better representations to generalize on new classes in this case. Roth et.al. (2020) also suggests Multisimilarity is a very strong baseline in deep metric learning which outperforms the proposed R-Margin on several datasets. And we use the hyperparameters of Multisimilarity recommended in Roth et.al. (2020) which generally perform well on multiple complex datasets. TinyImageNet gets much worse performance than other benchmarks because it has more classes (200), a longer task sequence (20 tasks), a larger feature space ($64 \times 64 \times 3$), and the accuracy of the single task on it is just about 17.8%. According to Tab. 3 the longer task sequence, more classes, and larger feature space all increase the gap between the performance of the single task and continual learning.

As shown in Tab. 6 the rho-spectrum shows high correlation to average accuracy on most benchmarks since it may help with learning new decision boundaries across tasks. Split MNIST has shown a low correlation between the $\rho$-spectrum and avg. accuracy due to the $\rho$-spectrum highly correlates with the avg. intransigence and consequently affect the avg. forgetting in an opposite direction so that causes a cancellation of effects on avg. accuracy. In addition, we found that GSS often obtains a smaller $\rho$ than other methods without getting a better performance. In general, the $\rho$-spectrum is the smaller the better because it indicates the representations are more informative. However, it may be detrimental to the performance when $\rho$ is too small as the learned representations are too noisy. DRL is more robust to this issue because $\rho$ keeps relatively stable when $\alpha$ is larger than a certain value as shown in Fig. 4c in Appx. E.

## 6    CONCLUSION

The two fundamental problems of continual learning with small episodic memories are: (*i*) how to make the best use of episodic memories; and (*ii*) how to construct most representative episodic memories. Gradient-based approaches have shown that the diversity of gradients computed on data from different tasks is a key to generalization over these tasks. In this paper we demonstrate that the

Table 3: Average accuracy (in %), the bold font indicates the best performance on this criterion

|  | P-MNIST | S-MNIST | Fashion | CIFAR10 | CIFAR100 | TinyImageNet |
|---|---|---|---|---|---|---|
| DRL | **78.7 ± 0.4** | **88.2 ± 0.6** | **78.2 ± 0.4** | 46.1 ± 1.2 | **17.1 ± 0.1** | **6.5 ± 0.4** |
| BER | 75.8 ± 0.3 | 86.4 ± 0.7 | 76.9 ± 0.6 | 44.2 ± 1.1 | 15.3 ± 0.6 | 5.7 ± 0.3 |
| ER | 76.1 ± 0.3 | 84.0 ± 0.8 | 75.6 ± 1.2 | 42.1 ± 2.0 | 14.5 ± 0.8 | 6.3 ± 0.2 |
| A-GEM | 75.9 ± 1.1 | 85.4 ± 0.8 | 60.6 ± 2.5 | 33.1 ± 2.1 | 9.8 ± 0.3 | 1.1 ± 0.3 |
| GSS | 77.1 ± 0.3 | 82.8 ± 1.8 | 72.5 ± 0.9 | 42.0 ± 3.0 | 13.9 ± 1.0 | 3.3 ± 0.2 |
| Multisim | 78.1 ± 0.2 | **88.1 ± 0.6** | 77.6 ± 0.5 | **49.5 ± 0.8** | 16.2 ± 0.3 | 6.3 ± 0.6 |
| R-Margin | 75.8 ± 0.4 | 86.0 ± 1.2 | 77.0 ± 0.6 | 46.0 ± 1.3 | 16.9 ± 0.5 | 6.0 ± 0.2 |
| Single | 82.3 ± 0.2 | 91.2 ± 0.3 | 80.5 ± 0.5 | 78.6 ± 2.1 | 33.5 ± 2.3 | 17.8 ± 0.4 |

Table 4: Average forgetting (in %), the bold font indicates the best performance on this criterion

|  | P-MNIST | S-MNIST | Fashion | CIFAR10 | CIFAR100 | TinyImageNet |
|---|---|---|---|---|---|---|
| DRL | 6.0 ± 0.3 | **8.4 ± 0.9** | 16.7 ± 1.5 | 32.2 ± 5.0 | 20.8 ± 0.9 | 9.4 ± 7.7 |
| BER | 7.1 ± 0.2 | 11.4 ± 1.0 | 17.4 ± 1.9 | 43.3 ± 2.1 | 20.6 ± 0.3 | **8.4 ± 0.3** |
| ER | 8.4 ± 0.3 | 15.6 ± 1.5 | 23.5 ± 1.7 | 48.6 ± 3.0 | 36.6 ± 0.5 | 32.6 ± 0.6 |
| A-GEM | **5.4 ± 1.1** | 10.7 ± 0.9 | 46.1 ± 3.4 | 34.7 ± 2.5 | **18.2 ± 1.1** | 15.8 ± 0.6 |
| GSS | 7.6 ± 0.2 | 17.9 ± 2.4 | 27.4 ± 2.2 | **10.9 ± 3.4** | 18.6 ± 0.7 | 11.3 ± 0.7 |
| Multisim | 5.9 ± 0.3 | 9.6 ± 0.9 | 18.3 ± 1.7 | 33.3 ± 2.1 | 25.9 ± 1.2 | 14.2 ± 0.4 |
| R-Margin | 6.9 ± 0.2 | 9.6 ± 1.4 | **14.0 ± 2.0** | 39.6 ± 4.5 | 24.6 ± 1.4 | 17.1 ± 0.4 |

Table 5: Average intransigence (in %), the bold font indicates the best performance on this criterion

|  | P-MNIST | S-MNIST | Fashion | CIFAR10 | CIFAR100 | TinyImageNet |
|---|---|---|---|---|---|---|
| DRL | **2.0 ± 0.1** | 2.6 ± 0.3 | 6.9 ± 1.0 | 12.1 ± 4.2 | 13.9 ± 0.8 | 25.0 ± 7.5 |
| BER | 4.0 ± 0.2 | 1.9 ± 0.2 | 7.7 ± 1.2 | 4.6 ± 1.2 | 15.9 ± 0.5 | 28.3 ± 0.5 |
| ER | 2.5 ± 0.1 | 0.9 ± 0.2 | 4.1 ± 0.6 | **2.6 ± 0.7** | **2.3 ± 0.7** | **2.7 ± 0.4** |
| A-GEM | 7.5 ± 0.5 | 3.1 ± 0.3 | **1.0 ± 0.3** | 22.7 ± 1.1 | 23.5 ± 1.0 | 24.1 ± 0.8 |
| GSS | 7.6 ± 0.2 | **0.9 ± 0.3** | 27.4 ± 2.2 | 35.7 ± 2.2 | 20.3 ± 1.4 | 38.6 ± 0.1 |
| Multisim | 2.6 ± 0.2 | 1.8 ± 0.3 | 6.2 ± 1.0 | 7.5 ± 1.1 | 10.3 ± 1.2 | 20.4 ± 0.7 |
| R-Margin | 4.1 ± 0.2 | 3.0 ± 0.3 | 10.5 ± 1.7 | 5.9 ± 2.6 | 10.7 ± 1.1 | 17.9 ± 0.3 |

Table 6: Correlation between model performance and $\rho$-spectrum on all benchmark tasks

| Coefficient | P-MNIST | S-MNIST | Split Fashion | CIFAR10 | CIFAR100 | TinyImageNet |
|---|---|---|---|---|---|---|
| Avg. Acc. | −0.8379 | 0.0461 | −0.5553 | −0.7689 | −0.7103 | −0.1820 |
| Avg. Forg. | 0.2616 | −0.3879 | 0.4331 | 0.1005 | 0.0028 | 0.1868 |
| Avg. Intran. | 0.4659 | 0.7206 | −0.0978 | 0.2463 | 0.2229 | −0.2377 |

most diverse gradients are from samples that are close to class boundaries. We formally connect the diversity of gradients to discriminativeness of representations, which leads to an alternative way to reduce the diversity of gradients in continual learning. We subsequently exploit ideas from DML for learning more discriminative representations, and furthermore identify the shared and different interests between continual learning and DML. In continual learning we would prefer larger margins between classes as the same as in DML. The difference is that continual learning requires less compact representations for better compatibility with future tasks. Based on these findings, we provide a simple yet efficient approach to solving the first problem listed above. Our findings also shed light on the second problem: it would be better for the memorized samples to preserve as much variance as possible. In most of our experiments, randomly chosen samples outperform those selected by gradient diversity (GSS) due to the limit on memory size in practice. It could be helpful to select memorized samples by separately considering the representativeness of inter- and intra-class samples, i.e., those representing margins and edges. We will leave this for future work.

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

## A  PROOF OF THEOREMS

**Notations**: Let $\mathcal{L}$ represent the softmax cross entropy loss, $\mathbf{W} \in \mathbb{R}^{D \times K}$ is the weight matrix of the linear model, and $\mathbf{x}_n \in \mathbb{R}^D$ denotes the input data, $\mathbf{y}_n \in \mathbb{R}^K$ is a one-hot vector that denotes the label of $\mathbf{x}_n$, $D$ is the dimension of representations, $K$ is the number of classes. Let $\boldsymbol{p}_n = softmax(\boldsymbol{o}_n)$, where $\mathbf{o}_n = \mathbf{W}^T \mathbf{x}_n$, the gradient $\boldsymbol{g}_n = \nabla_\mathbf{W} \mathcal{L}(\mathbf{x}_n, \mathbf{y}_n; \mathbf{W})$. $\mathbf{x}_n, \mathbf{x}_m$ are two different samples when $n \neq m$.

**Lemma 1.** *Let $\boldsymbol{\epsilon}_n = \boldsymbol{p}_n - \mathbf{y}_n$, we have $\langle \boldsymbol{g}_n, \boldsymbol{g}_m \rangle = \langle \mathbf{x}_n, \mathbf{x}_m \rangle \langle \boldsymbol{\epsilon}_n, \boldsymbol{\epsilon}_m \rangle$,*

*Proof.* Let $\boldsymbol{\ell}'_n = \partial \mathcal{L}(\mathbf{x}_n, \mathbf{y}_n; \mathbf{W}) / \partial \mathbf{o}_n$, by the chain rule, we have:

$$\langle \boldsymbol{g}_n, \boldsymbol{g}_m \rangle = \langle \mathbf{x}_n, \mathbf{x}_m \rangle \langle \boldsymbol{\ell}'_n, \boldsymbol{\ell}'_m \rangle,$$

By the definition of $\mathcal{L}$, we can find:

$$\boldsymbol{\ell}'_n = \boldsymbol{p}_n - \mathbf{y}_n, \tag{4}$$

$\square$

**Theorem 1.** *Suppose $\mathbf{y}_n \neq \mathbf{y}_m$, let $c_n$ denote the class index of $\mathbf{x}_n$ (i.e. $\mathbf{y}_{n,c_n} = 1, \mathbf{y}_{n,i} = 0, \forall i \neq c_n$). Let $\boldsymbol{\alpha} \triangleq ||\boldsymbol{p}_n||^2 + ||\boldsymbol{p}_m||^2$, $\boldsymbol{\beta} \triangleq \boldsymbol{p}_{n,c_m} + \boldsymbol{p}_{m,c_n}$ and $\boldsymbol{\delta} \triangleq ||\boldsymbol{p}_n - \boldsymbol{p}_m||_2^2$, then:*

$$\Pr(sign(\langle \boldsymbol{g}_n, \boldsymbol{g}_m \rangle) = sign(-\langle \mathbf{x}_n, \mathbf{x}_m \rangle)) = \Pr(2\boldsymbol{\beta} + \boldsymbol{\delta} > \boldsymbol{\alpha}),$$

*Proof.* According to Lemma 1 and $\mathbf{y}_n \neq \mathbf{y}_m$, we have

$$\langle \boldsymbol{\ell}'_n, \boldsymbol{\ell}'_m \rangle = \langle \boldsymbol{p}_n, \boldsymbol{p}_m \rangle - \boldsymbol{p}_{n,c_m} - \boldsymbol{p}_{m,c_n}$$

And

$$\langle \boldsymbol{p}_n, \boldsymbol{p}_m \rangle = \frac{1}{2}(||\boldsymbol{p}_n||^2 + ||\boldsymbol{p}_m||^2 - ||\boldsymbol{p}_n - \boldsymbol{p}_m||^2) = \frac{1}{2}(\boldsymbol{\alpha} - \boldsymbol{\delta})$$

which gives $\langle \boldsymbol{\ell}'_n, \boldsymbol{\ell}'_m \rangle = \frac{1}{2}(\boldsymbol{\alpha} - \boldsymbol{\delta}) - \boldsymbol{\beta}$. When $2\boldsymbol{\beta} > \boldsymbol{\alpha} - \boldsymbol{\delta}$, we must have $\langle \boldsymbol{\ell}'_n, \boldsymbol{\ell}'_m \rangle < 0$. According to Lemma 1, we prove this theorem. $\square$

**Theorem 2.** *Suppose $\mathbf{y}_n = \mathbf{y}_m$, when $\langle \boldsymbol{g}_n, \boldsymbol{g}_m \rangle \neq 0$, we have:*

$$sign(\langle \boldsymbol{g}_n, \boldsymbol{g}_m \rangle) = sign(\langle \mathbf{x}_n, \mathbf{x}_m \rangle),$$

*Proof.* Because $\sum_{k=1}^K \boldsymbol{p}_{n,k} = 1$, $\boldsymbol{p}_{n,k} \geq 0, \forall k$, and $c_n = c_m = c$,

$$\langle \boldsymbol{\ell}'_n, \boldsymbol{\ell}'_m \rangle = \sum_{k \neq c}^K \boldsymbol{p}_{n,k} \boldsymbol{p}_{m,k} + (\boldsymbol{p}_{n,c} - 1)(\boldsymbol{p}_{m,c} - 1) \geq 0 \tag{5}$$

According to Lemma 1, we prove the theorem. $\square$

## B  ALGORITHMS OF ONLINE MEMORY UPDATE

We provide the details of online ring buffer update and Balanced Experience Replay (BER) in Algs. 1 to 3. We directly load new data batches into the memory buffer without a separate buffer for the current task. The memory buffer works like a sliding window for each class in the data stream and we draw training batches from the memory buffer instead of directly from the data stream. In this case, one sample may not be seen only once as long as it stays in the memory buffer. This strategy is a more efficient use of the memory when $|\mathcal{B}| < n_c$, where $|\mathcal{B}|$ is the loading batch size of the data stream (i.e. the number of new samples added into the memory buffer at each iteration), we set $|\mathcal{B}|$ to 1 in all experiments (see Appx. I for a discussion of this).

**Algorithm 1** Ring Buffer Update with Fixed Buffer Size

---

**Input:** $\mathcal{B}_t$ - current data batch of the data stream, $\mathcal{C}_t$ - the set of classes in $\mathcal{B}_t$, $\mathcal{M}$ - memory buffer, $\mathcal{C}$ - the set of classes in $\mathcal{M}$, $K$ - memory buffer size.
**for** $c$ **in** $\mathcal{C}_t$ **do**
    Get $\mathcal{B}_{t,c}$ - samples of class $c$ in $\mathcal{B}_t$,
    $\mathcal{M}_c$ - samples of class $c$ in $\mathcal{M}$,
    **if** $c$ **in** $\mathcal{C}$ **then**
        $\mathcal{M}_c = \mathcal{M}_c \cup \mathcal{B}_c$
    **else**
        $\mathcal{M}_c = \mathcal{B}_c$,   $\mathcal{C} = \mathcal{C} \cup \{c\}$
    **end if**
**end for**
$R = |\mathcal{M}| + |\mathcal{B}| - K$
**while** $R > 0$ **do**
    $c' = \arg\max_c |\mathcal{M}_c|$
    remove the first sample in $\mathcal{M}_{c'}$,   $R = R - 1$
**end while**
return $\mathcal{M}$

**Algorithm 2** Balanced Experience Replay

---

**Input:** $\mathcal{M}$ - memory buffer, $\mathcal{C}$ - the set of classes in $\mathcal{M}$, $B$ - training batch size, $\Theta$ - model parameters, $\mathcal{L}_\Theta$ - loss function, $\mathcal{B}_t$ - current data batch from the data stream, $\mathcal{C}_t$ - the set of classes in $\mathcal{B}_t$, $K$ - memory buffer size.

$\mathcal{M} \leftarrow \text{MemoryUpdate}(\mathcal{B}_t, \mathcal{C}_t, \mathcal{M}, \mathcal{C}, K)$
$n_c, \mathcal{C}_s, \mathcal{C}_r \leftarrow \text{ClassSelection}(\mathcal{C}_t, \mathcal{C}, B)$
$\mathcal{B}_{train} = \emptyset$
**for** $c$ **in** $\mathcal{C}_s$ **do**
    **if** $c$ **in** $\mathcal{C}_r$ **then**
        $m_c = n_c + 1$
    **else**
        $m_c = n_c$
    **end if**
    Get $\mathcal{M}_c$ - samples of class $c$ in $\mathcal{M}$,
    $\mathcal{B}_c \overset{m_c}{\sim} \mathcal{M}_c \lhd$ sample $m_c$ samples from $\mathcal{M}_c$
    $\mathcal{B}_{train} = \mathcal{B}_{train} \cup \mathcal{B}_c$
**end for**
$\Theta \leftarrow \text{Optimizer}(\mathcal{B}_{train}, \Theta, \mathcal{L}_\Theta)$

**Algorithm 3** Class Selection for BER

---

**Input:** $\mathcal{C}_t$ - the set of classes in current data batch $\mathcal{B}_t$, $\mathcal{C}$ - the set of classes in $\mathcal{M}$, $B$ - training batch size, $m_p$ - minimum number of positive pairs of each selected class ($m_p \in \{0,1\}$).
$\mathcal{B}_{train} = \emptyset, n_c = \lfloor B/|\mathcal{C}| \rfloor, \; r_c = B \mod |\mathcal{C}|$,
**if** $n_c > 1$ or $m_p == 0$ **then**
    $\mathcal{C}_r \overset{r_c}{\sim} \mathcal{C}$    $\lhd$ sample $r_c$ classes from all seen classes without replacement.
    $\mathcal{C}_s = \mathcal{C}$
**else**
    $\mathcal{C}_r = \emptyset, n_c = 2, n_s = \lfloor B/2 \rfloor - |\mathcal{C}_t|, \lhd$ we ensure the training batch include samples from the current task.
    $\mathcal{C}_s \overset{n_s}{\sim} (\mathcal{C} - \mathcal{C}_t) \lhd$ sample $n_s$ classes from all seen classes except classes in $\mathcal{C}_t$.
    $\mathcal{C}_s = \mathcal{C}_s \bigcup \mathcal{C}_t$
    **if** $B \mod 2 > 0$ **then**
        $\mathcal{C}_r \overset{1}{\sim} \mathcal{C}_s$    $\lhd$ sample one class in $\mathcal{C}_s$ to have an extra sample.
    **end if**
**end if**
**Return:** $n_c, \mathcal{C}_s, \mathcal{C}_r$

## C  DEFINITION OF PERFORMANCE MEASURES

We use the following measures to evaluate the performance of all methods:

*Average accuracy*, which is evaluated after learning all tasks: $\bar{a}_t = \frac{1}{t} \sum_{i=1}^{t} a_{t,i}$, where $t$ is the index of the latest task, $a_{t,i}$ is the accuracy of task $i$ after learning task $t$.

*Average forgetting* (Chaudhry et al., 2018), which measures average accuracy drop of all tasks after learning the whole task sequence: $\bar{f}_t = \frac{1}{t-1} \sum_{i=1}^{t-1} \max_{j \in \{i,...,t-1\}} (a_{j,i} - a_{t,i})$.

*Average intransigence* (Chaudhry et al., 2018), which measures the inability of a model learning new tasks: $\bar{I}_t = \frac{1}{t} \sum_{i=1}^{t} a_i^* - a_i$, where $a_i$ is the accuracy of task $i$ at time $i$. We use the best accuracy among all compared models as $a_i^*$ instead of the accuracy obtained by an extra model that is solely trained on task $i$.

# D    RELATED METHODS FROM DML

$\rho$-**spectrum metric** (Roth et al., 2020): $\rho = KL(\mathcal{U}||S_{\Phi_{\mathcal{X}}})$, which is proposed to measure the information entropy contained in the representation space. The $\rho$-spectrum computes the KL-divergence between a discrete uniform distribution $\mathcal{U}$ and the spectrum of data representations $S_{\Phi_{\mathcal{X}}}$, where $S_{\Phi_{\mathcal{X}}}$ is normalized and sorted singular values of $\Phi(\mathcal{X})$ , $\Phi$ denotes the representation extractor (e.g. a neural network) and $\mathcal{X}$ is input data samples. Lower values of $\rho$ indicate higher variance of the representations and hence more information entropy retained.

**Multisimilarity**(Wang et al., 2019): we adopt the loss function of Multisimilarity as an auxiliary objective in classfication tasks of continual learning, the batch mining process is omitted because we use labels for choosing positive and negative pairs. So the loss function is $\widehat{\mathcal{L}} = \mathcal{L} + \lambda\mathcal{L}_{multi}$, and:

$$\mathcal{L}_{multi} = \frac{1}{B}\sum_{i=1}^{B}\left[\frac{1}{\alpha}\log[1 + \sum_{j\neq i, y_j = y_i}\exp\left(-\alpha(s_c(h_i, h_j) - \gamma)\right)]\right.$$
$$\left. + \frac{1}{\beta}\log\left[1 + \sum_{y_j \neq y_i}\exp\left(\beta(s_c(h_i, h_j) - \gamma)\right)\right]\right] \tag{6}$$

where $s_c(\cdot, \cdot)$ is cosine similarity, $\alpha, \beta, \gamma$ are hyperparameters. In all of our experiments we set $\alpha = 2, \beta = 40, \gamma = 0.5$ as the same as in Roth et al. (2020).

**R-Margin**(Roth et al., 2020): we similarly deploy R-Margin for continual learning as the above, which uses the Margin loss (Wu et al., 2017) with the $\rho$ regularization (Roth et al., 2020) as introduced in Sec. 2.2. So the loss function is $\widehat{\mathcal{L}} = \mathcal{L} + \lambda\mathcal{L}_{margin}$, and:

$$\mathcal{L}_{margin} = \sum_{i=1}^{B}\sum_{j=1}^{B}\gamma + \mathbb{I}_{j\neq i, y_j = y_i}(d(h_i, h_j) - \beta) - \mathbb{I}_{y_j \neq y_i}(d(h_i, h_j) - \beta) \tag{7}$$

where $d(\cdot, \cdot)$ is Euclidean distance, $\beta$ is a trainable variable and $\gamma$ is a hyperparameter. We follow the setting in Roth et al. (2020): $\gamma = 0.2$, the initialization of $\beta$ is 0.6. We set $p_\rho = 0.2$ in $\rho$ regularization.

# E    ABLATION STUDY ON DRL

We verify the effects of $\mathcal{L}_{DRL}$ by training a model with/without $\mathcal{L}_{DRL}$ on Split-MNIST tasks: Fig. 4a shows that $\mathcal{L}_{DRL}$ notably reduces the similarities of representations from different classes while making representations from a same class less similar; Fig. 4b shows the analogous effect on gradients from different classes and a same class. Fig. 4c demonstrates increasing $\alpha$ can effectively decrease $\rho$-spectrum to a low-value level, where lower values of $\rho$ indicate higher variance of the representations and hence more information entropy retained.

Tab. 7 provides the results of an ablation study on the effects of the two terms in DRL. In general, $\mathcal{L}_{bt}$ gets a better performance in terms of forgetting, $\mathcal{L}_{wi}$ gets a better performance in terms of intransigence and a lower $\rho$-spectrum, and both of them show improvements on BER (without any regularization terms). Overall, combining the two terms obtains a better performance on forgetting than standalone $\mathcal{L}_{bt}$ and keeps the advantage on intransigence that brought by $\mathcal{L}_{wi}$. It indicates preventing over-compact representations while maximizing margins can improve the learned representations that are easier for generalization over previous and new tasks. In addition, we found that using standalone $\mathcal{L}_{bt}$ we can only use a smaller $\lambda$ otherwise the gradients will explode, and using $\mathcal{L}_{wi}$ together can stablize the gradients.

We notice that the lower $\rho$-spectrum does not necessarily lead to a higher accuracy as it's correlation coefficients with accuracy depends on datasets and is usually larger than -1.

# F    COMPARING DIFFERENT MEMORY SIZES

Fig. 5 compares average accuracy of DRL+BER on MNIST tasks with different memory sizes. It appears the fixed memory size is more efficient than the incremental memory size. For example, the

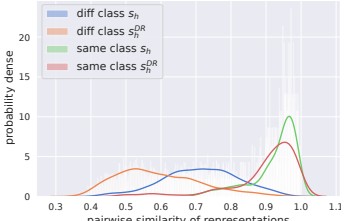 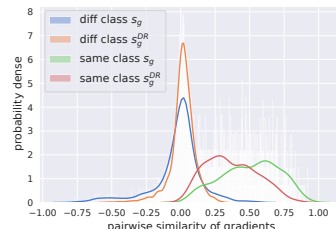 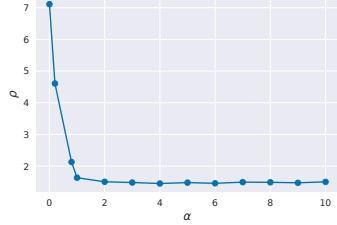

(a) Similarities of representations with and without $\mathcal{L}_{DRL}$

(b) Similarities of gradients with and without $\mathcal{L}_{DRL}$

(c) Relation between $\alpha$ and $\rho$-spectrum.

Figure 4: Effects of $\mathcal{L}_{DRL}$ on reducing diveristy of gradients and $\rho$-spectrum. (a) and (b) display distributions of similarities of representations and gradients. $s_h^{DR}$ and $s_h$ denote similarities of representations with and without $\mathcal{L}_{DRL}$, respectively, $s_g^{DR}$ and $s_g$ denote similarities of gradients with and without $\mathcal{L}_{DRL}$, respectively. (c) demonstrates increasing $\alpha$ in $\mathcal{L}_{DRL}$ can reduce $\rho$ effectively.

Table 7: Comparing the performance with or without the regularization terms ($\mathcal{L}_{bt}$, $\mathcal{L}_{wi}$) in DRL. All criteria are in % except $\rho$-spectrum. The bold font inditactes the best performance of a criterion, as $\rho$-spectrum is not a performance measurement so we do not put bold font on it.

|  |  | BER | DRL | | |
|---|---|---|---|---|---|
|  |  | with none | with $\mathcal{L}_{bt}$ | with $\mathcal{L}_{wi}$ | with both |
| P-MNIST | Avg. Accuracy | $75.8 \pm 0.3$ | $77.0 \pm 0.3$ | $77.8 \pm 0.4$ | $\mathbf{78.7 \pm 0.4}$ |
|  | Avg. Forgetting | $7.1 \pm 0.2$ | $6.7 \pm 0.3$ | $6.6 \pm 0.4$ | $\mathbf{6.0 \pm 0.3}$ |
|  | Avg. Intransigence | $4.0 \pm 0.2$ | $3.1 \pm 0.2$ | $2.3 \pm 0.3$ | $\mathbf{2.0 \pm 0.1}$ |
|  | $\rho$-spectrum | $1.11 \pm 0.09$ | $0.72 \pm 0.02$ | $0.65 \pm 0.02$ | $0.71 \pm 0.01$ |
| S-MNIST | Avg. Accuracy | $86.4 \pm 0.7$ | $87.3 \pm 1.0$ | $87.4 \pm 1.1$ | $\mathbf{88.2 \pm 0.6}$ |
|  | Avg. Forgetting | $11.4 \pm 1.0$ | $10.0 \pm 1.3$ | $10.6 \pm 1.5$ | $\mathbf{8.4 \pm 0.9}$ |
|  | Avg. Intransigence | $2.6 \pm 0.3$ | $2.8 \pm 0.2$ | $\mathbf{2.3 \pm 0.2}$ | $2.7 \pm 0.2$ |
|  | $\rho$-spectrum | $2.07 \pm 0.16$ | $1.35 \pm 0.2$ | $1.07 \pm 0.06$ | $1.05 \pm 0.03$ |
| Fashion | Avg. Accuracy | $76.9 \pm 0.6$ | $77.3 \pm 1.0$ | $77.2 \pm 1.1$ | $\mathbf{78.2 \pm 0.4}$ |
|  | Avg. Forgetting | $17.4 \pm 1.9$ | $17.2 \pm 2.3$ | $18.4 \pm 2.2$ | $\mathbf{16.7 \pm 1.5}$ |
|  | Avg. Intransigence | $7.7 \pm 1.2$ | $7.1 \pm 1.3$ | $\mathbf{6.6 \pm 1.2}$ | $6.9 \pm 1.0$ |
|  | $\rho$-spectrum | $2.07 \pm 0.21$ | $1.55 \pm 0.09$ | $1.22 \pm 0.06$ | $1.21 \pm 0.04$ |

fixed memory size (M = 300) getting very similar average accuracy with memory M = 50 per class in Disjoint MNIST while it takes less cost of the memory after task 3. Meanwhile, the fixed memory size (M = 300) gets much better performance than M = 50 per task in most tasks of Permuted MNIST and it takes less cost of the memory after task 6. Since the setting of fixed memory size takes larger memory buffer in early tasks, the results indicate better generalization of early tasks can benefit later tasks, especially for more homogeneous tasks such as Permuted MNIST. The results also align with findings about Reservoir sampling (which also has fixed buffer size) in Chaudhry et al. (2019b) and we also believe a hybrid memory strategy can obtain better performance as suggested in Chaudhry et al. (2019b).

## G COMPARING DIFFERENT REPLAY STRATEGY

We compare DRL with different memory replay strategies in Tab. 8 to show DRL has general improvement based on the applied replay strategy.

## H COMPARING TRAINING TIME

Tab. 9 compares the training time of MNIST tasks. All representation-based methods are much faster than gradient-based methods and close to the replay-based methods.

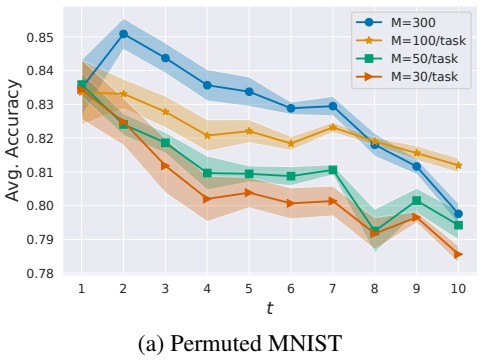
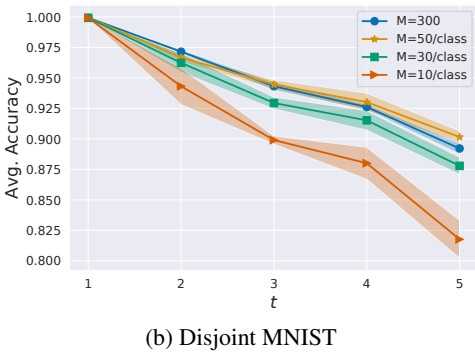

| (a) Permuted MNIST | (b) Disjoint MNIST |
|---|---|

Figure 5: Average accuracy of DRL+BER with different memory sizes. The $x$ axis is the index of tasks, the shaded area is plotted by standard deviation of 5 runs.

Table 8: Comparing DRL with different memory replay strategies, all criteria are in %.

| | | DRL+BER | DRL+ER | BER | ER |
|---|---|---|---|---|---|
| | P-MNIST | $78.7 \pm 0.4$ | $76.9 \pm 0.5$ | $75.8 \pm 0.3$ | $76.1 \pm 0.3$ |
| Avg. Accuracy | S-MNIST | $88.2 \pm 0.6$ | $85.9 \pm 0.8$ | $86.4 \pm 0.7$ | $84.0 \pm 0.8$ |
| | Fashion | $78.2 \pm 0.4$ | $76.6 \pm 1.2$ | $76.9 \pm 0.6$ | $75.6 \pm 1.2$ |
| | P-MNIST | $6.0 \pm 0.3$ | $6.0 \pm 0.6$ | $7.1 \pm 0.2$ | $8.4 \pm 0.3$ |
| Avg. Forgetting | S-MNIST | $8.4 \pm 0.9$ | $13.4 \pm 1.1$ | $11.4 \pm 1.0$ | $15.6 \pm 1.5$ |
| | Fashion | $16.7 \pm 1.5$ | $22.7 \pm 1.5$ | $17.4 \pm 1.9$ | $23.5 \pm 1.7$ |
| | P-MNIST | $1.2 \pm 0.2$ | $2.9 \pm 0.2$ | $3.1 \pm 0.2$ | $1.6 \pm 0.2$ |
| Avg. Intransigence | S-MNIST | $2.6 \pm 0.3$ | $1.2 \pm 0.3$ | $1.9 \pm 0.2$ | $0.9 \pm 0.2$ |
| | Fashion | $6.9 \pm 1.0$ | $5.4 \pm 0.4$ | $7.7 \pm 1.2$ | $4.1 \pm 0.6$ |

Table 9: Training time (in seconds) of the whole task sequence of MNIST tasks, which have been tested on a laptop with an 8-core Intel CPU and 32G RAM.

| | DRL | BER | ER | A-GEM | GSS | Multisim | R-Margin |
|---|---|---|---|---|---|---|---|
| Permuted | $12.48 \pm 0.16$ | $11.17 \pm 0.18$ | $\mathbf{10.38 \pm 0.05}$ | $28.0 \pm 0.09$ | $33.98 \pm 0.6$ | $12.91 \pm 0.13$ | $13.45 \pm 0.14$ |
| Split | $5.6 \pm 0.14$ | $5.29 \pm 0.08$ | $\mathbf{5.25 \pm 0.02}$ | $13.41 \pm 0.47$ | $19.07 \pm 1.31$ | $5.89 \pm 0.09$ | $6.29 \pm 0.43$ |

## I HYPER-PARAMETERS IN EXPERIMENTS

To make a fair comparison of all methods, we use following settings: *i*) The configurations of GSS-greedy are as suggested in Aljundi et al. (2019), with batch size set to 10 and each batch receives 10 iterations. *ii*) For the other methods, we use the ring buffer memory as described in Alg. 1, the loading batch size is set to 1, following with one iteration, the training batch size is provided in Tab. 10. More hyperparameters are given in Tab. 10 as well.

In the setting of limited training data in online continual learning, we either use a small batch size or iterate on one batch several times to obtain necessary steps for gradient optimization. We chose a small batch size with one iteration instead of larger batch size with multiple iterations because by our memory update strategy (Alg. 1) it achieves similar performance with fewer hyperparameters. Since GSS-greedy has a different strategy for updating memories, we leave it at its default settings.

Regarding the two terms in DRL, a larger weight on $\mathcal{L}_{wi}$ is for less compact representations within classes, but a too dispersed representation space may include too much noise. For datasets that present more difficulty in learning compact representations, we would prefer a smaller weight on $\mathcal{L}_{wi}$, we therefore set smaller $\alpha$ for CIFAR datasets in our experiments. A larger weight on $\mathcal{L}_{bt}$ is more resistant to forgetting but may be less capable of transferring to a new task, for datasets that are less compatible between tasks a smaller weight on $\mathcal{L}_{bt}$ would be preferred, as we set the largest $\lambda$ on Permuted MNIST and the smallest $\lambda$ on CIFAR-100 in our experiments.

Table 10: Hyperparameters of all experiments

|  | P-MNIST | S-MNIST | Fashion | CIFAR-10 | CIFAR-100 | TinyImageNet |
|---|---|---|---|---|---|---|
| training batch size | 20 | 20 | 20 | 10 | 10 | 20 |
| learning rate | 0.02 | 0.2 | 0.2 | 0.1 | 0.2 | 0.2 |
| ref batch size (A-GEM) | 256 | 256 | 256 | 256 | 1500 | 1500 |
| $\alpha$ of DRL | 2 | 2 | 2 | 0.1 | 0.1 | 0.1 |
| $\lambda$ of DRL | $1 \times 10^{-2}$ | $5 \times 10^{-4}$ | $5 \times 10^{-4}$ | $2 \times 10^{-4}$ | $2 \times 10^{-5}$ | $2 \times 10^{-5}$ |
| $\lambda$ of Multisim | 5 | 1 | 1 | 2 | 0.1 | 0.1 |
| $\lambda$ of R-Margin | $2 \times 10^{-5}$ | $1 \times 10^{-3}$ | $1 \times 10^{-3}$ | $1 \times 10^{-4}$ | $2 \times 10^{-4}$ | $2 \times 10^{-4}$ |

Table 11: The grid search range of Hyperparameters of all experiments except TinyImageNet, as we use the same hyperparameters of CIFAR-100 to TinyImageNet except increasing the training batch size to 20.

|  | The grid-search range |
|---|---|
| training batch size | [10, 20, 50, 100] |
| learning rate | [0.001, 0.01, 0.02, 0.1, 0.2] |
| ref batch size (A-GEM) | [128, 256, 512, 1000, 1500, 2000] |
| $\alpha$ of DRL | [0.1, 0.2, 0.5, 1, 2, 4] |
| $\lambda$ of DRL | $[1 \times 10^{-5}, 2 \times 10^{-5}, 5 \times 10^{-5}, 1 \times 10^{-4}, 2 \times 10^{-4}, 5 \times 10^{-4}, 1 \times 10^{-3}, 1 \times 10^{-2}, 1 \times 10^{-1}]$ |
| $\lambda$ of Multisim | [10, 8, 6, 5, 4, 3, 2, 1, 0.5, 0.2, 0.1, 0.05] |
| $\lambda$ of R-Margin | $[1 \times 10^{-5}, 2 \times 10^{-5}, 5 \times 10^{-5}, 1 \times 10^{-4}, 2 \times 10^{-4}, 5 \times 10^{-4}, 1 \times 10^{-3}, 1 \times 10^{-2}, 1 \times 10^{-1}]$ |

