# OpenReview forum: "Discriminative Representation Loss (DRL): A More Efficient Approach than Gradient Re-Projection in Continual Learning"
_ICLR.cc/2021/Conference — Reject_

### Official Review · AnonReviewer1 · 2020-10-28
**Insufficient experiments**

**Rating:** 6
**Confidence:** 3

**Review:**

This paper presents a novel way of making full use of compact episodic memory to alleviate catastrophic forgetting in continual learning. This is done by adding the proposed discriminative representation loss to regularize the gradients produced by new samples. Authors gave insightful analysis on the influence of gradient diversity to the performance of continual learning, and proposed a regularization that connects metric learning and continual learning. However, there are still some issues to be addressed as below.

1. Authors do not explain how to determine weights of the two terms in the proposed discriminative representation loss to achieve good balance between discrmination capability on the current task and generalization on future task.

2. No experiment on the influence of memory size and batch size on the proposed method. These are important hyperparameters for the proposed method. So there should be experiment inspecting its influnece on the performance.

3. There is also no experiment on the influence of different memory update rule on the proposed method. It is not clear how well the proposed method is robust to different memory update rules.

4. Experiments on more diverse datasets should be done to prove its effectiveness, such as TinyImageNet, and MIT Scenes/Oxford Flowers/UCSD Birds/Stanford Cars.

---

> ### Author Response · Authors · 2020-11-20
> **Response to R1's comments**
>
> We really appreciate R1 for recognizing the contribution of our work and will add more experiments in the revision. In the following we address R1’s main concerns one by one.
>
> “Authors do not explain how to determine weights of the two terms in the proposed discriminative representation loss to achieve good balance between discrmination capability on the current task and generalization on future task.”
>
> Regarding the two terms in DRL, a larger weight on L_wi is for less compact representations within classes, but a too dispersed representation space may include too much noise. For datasets that present more difficulty in learning compact representations, we would prefer a smaller weight on L_wi, we therefore set smaller \alpha for CIFAR datasets in our experiments. A larger weight on L_bt is more resistant to forgetting but may be less capable of transferring to a new task, for datasets that are less compatible between tasks a smaller weight on L_bt would be preferred, as we set the largest \lambda on Permuted MNIST and the smallest \lambda on CIFAR-100 in our experiments.  Table 6 in the appendix shows the hyperparamters used in all the experiments.
>
> “No experiment on the influence of memory size and batch size on the proposed method. These are important hyperparameters for the proposed method. So, there should be experiment inspecting its influnece on the performance.”
>
> The memory size is the larger the better for performance which is consistent in the literature of continual learning and a perfect memory would lead to a perfect performance as suggested in Knoblauch et al. (2020).  We provide experiment results of comparing different memory sizes in the Appendix E in the revision. Most of the compared baselines use a small batch size in their original papers (such as 10 in A-GEM & GSS, 20 in ER), and we did not find large batch size can improve the performance, also the batch size highly relates to the computational complexity of most methods, so we keep it small in all experiments.
>
> Knoblauch, Jeremias, Hisham Husain, and Tom Diethe. "Optimal Continual Learning has Perfect Memory and is NP-hard." ICML (2020).
>
> “There is also no experiment on the influence of different memory update rule on the proposed method. It is not clear how well the proposed method is robust to different memory update rules.”
>
> The performance difference between DRL+ ER and DRL+BER is similar as the difference between ER and BER, DRL+ER is better than ER but worse than DRL+BER.  We included results of comparing DRL with different replay strategies in Appendix F in the revision.
>
> “Experiments on more diverse datasets should be done to prove its effectiveness, such as TinyImageNet, and MIT Scenes/Oxford Flowers/UCSD Birds/Stanford Cars.”
>
> We provided experimental results on TinyImageNet in the revision, in which DRL outperforms other baselines.

---

### Official Review · AnonReviewer3 · 2020-10-29

**Rating:** 6
**Confidence:** 4

**Review:**

This paper analyzes the relationship of gradient diversity to the performance of continual learning systems, the analysis inspires a novel loss which shows some improvement in the continual learning setting.

Strengths:

-The empirical analysis is interesting making connections to Liu 2020 and some effective visualizations. The linear cases is studied in detail and provides intuition for the method

-The proposed loss gives some improvement in larger memory settings


Weaknesses:

-Proposed method is too informally motivated by diversity analysis. It’s not clear to the reviewer how the theoretical observation regarding the linear case can be extended to the non-linear case.

-Experimental results are promising but not completely convincing especially with the 2 additional hyperparameters:
1. The authors use buffer settings much higher than prior works (e.g. [a,b] consider memories of 200,500,1000), this choice is not explained nor lack of comparison to any existing published result
2. There are several works e.g. [a,b,c] which consider the same setting as this work and have better performance than the baselines shown. For example for CIFAR-10 [a] uses smaller memory size and has higher performance than shown here.

-Cross entropy is already indirectly bounding something similar to the L_bt-L_wi ([e]), so the objective with 2 hyper-parameters seems overly cumbersome.

-Related Work is a bit sparse. Besides the works mentioned above there is several considering metric learning based approaches for continual learning which should be discussed (e.g. [d])

Question:
-Are the hidden representations normalized?
-The minibatches used are small and thus can have few same class pairs, how do the authors assure that L_wi can be optimized in this case, particularly with say 100 classes as in CIFAR-100.

Overall, I find the work promising. The observations regarding gradient diversity might be the basis of future more effective methods. On the other hand the exact method proposed and results are not completely convincing in their current state.

[a] Aljundi et al “Online Continual Learning with Maximally Interfered Retrieval”
[b] Ji et al “Automatic recall machines: Internal replay, continual learning and the brain”
[c] Caccia et al “Online Learned Continual Compression with Adaptive Quantization Modules”
[d]  Li et al  “Better Knowledge Retention through Metric Learning”
[e] Boudiaf "A unifying mutual information view of
metric learning: cross-entropy vs. pairwise losses"

----------------

Post-rebuttal:
After reading the rebuttal I maintain my score. The observations seem promising, the method a bit cumbersome but also interesting, but neither of them are fully fleshed out. The authors should either greatly expand the empirical analysis (in the non-linear setting) of their claims on intra and inter class variability in CL  and/or make the experiments of the DRL method more convincing and varied in scope

---

> ### Author Response · Authors · 2020-11-20
> **Response to R3's comments**
>
> We really appreciate R3 for recognizing the contribution of our work and will clarify most of the issues in the revision. In the following we address R3’s main concerns one by one.
>
> “-Proposed method is too informally motivated by diversity analysis. It’s not clear to the reviewer how the theoretical observation regarding the linear case can be extended to the non-linear case.”
>
> The Theorems explain in which cases we are likely to get negative inner product of gradients, which gives two conditions: 1). positive inner product of inter-class representations with unconfident predictions; 2) negative inner product of within-class representations. It indicates decreasing the similarity of inter-class representations can be helpful with reducing negative inner product of gradients. We empirically verify this on a non-linear model by the experiments in Fig.2, which demonstrates smaller representation similarity of inter-class samples leads to less negative gradient similarity.  We provide more empirical study of connecting the non-linear case to the linear case in Sec.2 in the revision, which shows the non-linear model generally behaves very similar with linear model and aligns with the theorems.
>
> “-Experimental results are promising but not completely convincing especially with the 2 additional hyperparameters:
> The authors use buffer settings much higher than prior works (e.g. [a,b] consider memories of 200,500,1000), this choice is not explained nor lack of comparison to any existing published result.
> There are several works e.g. [a,b,c] which consider the same setting as this work and have better performance than the baselines shown. For example, for CIFAR-10 [a] uses smaller memory size and has higher performance than shown here.”
>
> We tested a fixed memory size for a fair comparison with GSS because it does not select samples by class index when forming the episodic memory. Regarding the experiments of CIFAR-10, the training size per task is 9750 samples and the network is a standard ResNet18 in [a,b,c], whereas in our settings, the training size is 3000 samples per task and the network is a reduced ResNet18 as the same as Chaudhry et.al. (2019a), which increases the difficulty of this benchmark and shows data efficiency of our method.  Referring to the experiments of split-MNIST in [a], they use a larger network with 400 hidden units per layer whereas we use 100, and they use a memory with 50 samples per class whereas we use 300 in total (30 per class in the final task), and our method gives a better avg. accuracy.
>
> “-Cross entropy is already indirectly bounding something similar to the L_bt-L_wi ([e]), so the objective with 2 hyper-parameters seems overly cumbersome.”
>
> The objective of DRL is L_bt + \alpha L_wi, in which the L_wi has an opposite sign of the suggested form in [e], - L_wi  corresponds to learning compact representations within classes, and in our method, +L_wi corresponds to preventing over compactness within classes.  So \alpha controls an opposite strength of L_wi which is not bounded by cross entropy.
>
> “Question: -Are the hidden representations normalized? -The minibatches used are small and thus can have few same class pairs, how do the authors assure that L_wi can be optimized in this case, particularly with say 100 classes as in CIFAR-100.”
>
> The hidden representations are normalized in Multisimilarity and R-Margin as it is explicitly required in the two papers. They are not normalized in DRL as we found there is no significant difference without normalization. We guarantee the positive pairs in each batch as described in the last three sentences of Sec. 4 and Alg. 2 in the Appendix B.

---

> > ### Comment · AnonReviewer3 · 2020-11-22
> > **Question**
> >
> > Thank you for your reply.
> >
> > The authors have made a reasonable summary about the link of the linear to the non-linear case. I encourage this sort of concise explanation to be incorporated in the text. However as noted by R5 the experimental evidence in the non-linear case is still limited, particularly the claims about within class similarity need to be further explored.
> >
> > Regarding the  experiments in Sec 5, it is still not clear to the reviewer why settings matching those in prior works are not selected. For example the authors state they use 3000 samples in the CIFAR-10 experiments unlike [a,b,c] and other minor differences like width of NN are highlighted but why these couldnt be simply follow the prior work? Switching to 1/3 of the data is hardly a limited data setting and furthermore limited data is not discussed anywhere else. Also in looking at [a] it seems to use reduced ResNet-18 as in Chaudhry et al unlike stated above.  The authors have restated the differences but not clarified why they didnt compare to existing settings and why their specific modifications of the other settings are more appropriate here.

---

> > > ### Author Response · Authors · 2020-11-23
> > > **Response to R3's remaining concerns**
> > >
> > > We thank R3’s timely feedback to our response and we address the remaining concerns as the following:
> > >
> > >
> > >
> > > “The authors have made a reasonable summary about the link of the linear to the non-linear case. I encourage this sort of concise explanation to be incorporated in the text. However as noted by R5 the experimental evidence in the non-linear case is still limited, particularly the claims about within class similarity need to be further explored.”
> > >
> > > We added more analysis about the within class loss in Sec. 2.2 in the revision. The experimental results in Fig.1 also indicate that we should not learn too compact representations in a current task which might omit important dimensions in the representation space for a future task, e.g., omitting the y-dimension would not affect task 1 but would affect task 2. In this case, even if we store diverse samples into the memory, the learned representations may be insufficient to generalize on future tasks as the omitted dimensions can only be relearned by using limited samples in the memory. We demonstrate this through experiments with and without L1 regularization in a MLP for the first two tasks of split-MNIST and split-Fashion MNIST, the results are provided in Tab.2, Sec. 2.2 in the revision. And we also provide an ablation study on the two terms of DRL in Appendix E in the revision.
> > >
> > >
> > >
> > > “Regarding the experiments in Sec 5, it is still not clear to the reviewer why settings matching those in prior works are not selected. For example the authors state they use 3000 samples in the CIFAR-10 experiments unlike [a,b,c] and other minor differences like width of NN are highlighted but why these couldnt be simply follow the prior work? Switching to 1/3 of the data is hardly a limited data setting and furthermore limited data is not discussed anywhere else. Also in looking at [a] it seems to use reduced ResNet-18 as in Chaudhry et al unlike stated above. The authors have restated the differences but not clarified why they didnt compare to existing settings and why their specific modifications of the other settings are more appropriate here.”
> > >
> > > We use 100 units per layer in the MLP which matches the settings in GSS (Aljundi et al., 2019) and GEM (Lopez-Paz et al., 2017)). In GSS the experiments of CIFAR-10 use 2000 samples per task with 1000 memory size, but we found that by using this setting  A-GEM is difficult to get reasonable results, so to be fair to their method we increased the training size and memory size.  In [a], it is stated in the Appendix B.1 that they use a standard Resnet-18 as in GEM and A-GEM, but we weren’t able to find references to GEM using a standard Resnet-18. We did note that A-GEM uses a standard Resnet-18 for CUB and AWA, and reduced Resnet-18 for CIFAR, so this is perhaps the source of the confusion. Overall, we tried to ensure that the setup was as fair to all of the methods as possible within the computational constraints that we faced.

---

### Official Review · AnonReviewer5 · 2020-11-06
**Connecting representation learning to continual learning would be of great value, but the theoretical and empirical claims made in the paper are insufficiently substantiated**

**Rating:** 5
**Confidence:** 4

**Review:**

############## Summary ##############

This submission draws a connection between deep representation learning and continual learning. The authors include theoretical and empirical analyses that suggest that the learned model for continual learning should both separate the representations of instances of different classes and separate the representations of instances within a single class. This motivates a new representation-learning-based continual learning method called DRL, which the authors evaluate empirically against various baselines.

############## Strengths ##############

1. The problem studied in this work is interesting. A deeper understanding of what types of representations could enable continual learning could potentially be very impactful to the field.
2. The paper includes both theoretical and empirical analyses.
3. The experimental setting is described in detail and includes a comprehensive set of baselines.

############## Weaknesses ##############

1. The theoretical findings seem to be somewhat disconnected from the empirical study in Section 2.1, and the analysis connecting the two is unconvincing. Moreover, given this disconnect, the empirical study should have been much more complete, including various data sets.
2. The writing and the structure of the paper is quite unclear, leaving the reader confused at times, having to jump back and forth to draw connections between different sections that are not explicitly described in the text.
3. The experimental evaluation is inconsistent in showing the advantage of their method, and this is not analyzed in any depth.


############## Recommendation ##############

Unfortunately, I lean towards recommending the rejection of this paper. While the studied problem is of high relevance, whose answer could be very impactful, I find the theoretical and empirical analyses in this work to be unconvincing, and thus the claims made throughout the paper to be insufficiently founded. The authors use very simple toy experiments and theorems on linear models to motivate their claim that learned representations should be very different across different classes. While using theory from linear models is a common technique for motivating empirical study of deep learning models, a much more comprehensive empirical study would be needed to validate that the theoretical findings extend to the deep learning setting. Moreover, the paper should be substantially revised both to add clarity to the motivation of the problem, and to more carefully analyze the connections between the theoretical and empirical analyses in Section 2.1.


############## Arguments ##############

The biggest reason I lean towards rejecting this paper is that I am unconvinced by the claims made throughout Section 2.

First, the section begins by suggesting that instances with similar gradients are those most useful for generalization. This seems to contradict the findings of Aljundi et al., which chooses to keep samples with maximum gradient diversity in the episodic memory. This apparent contradiction should be carefully discussed and analyzed.

Then, the experiment of Figure 1 suggests that: the gradients with the most diverse gradients are those close to the decision boundary, and in order to avoid forgetting previously seen tasks, samples with diverse representations help maintain good decision boundaries as new tasks are learned. The former claim is further studied in the rest of the section, but why does this experiment suggest that we would like to keep samples with diverse gradients? I keep seeing conflicting indications that we should store samples with diverse or similar gradients. On the other hand, the latter claim is not substantiated beyond these toy experiments. It seems to be specific to incremental-class learning, and would probably not hold for other variants of continual learning that don't require the same model to discern between the new class and the previous ones.

In Theorem 1, the authors claim that points close to the decision boundary are likely to yield negative inner product of their gradients. Figure 4 in the Appendix seeks to illustrate this, and could potentially be very helpful towards understanding this work better, but  1) the figure is missing a y-axis label and 2) the caption is confusing and contradictory between the sub-figure captions and the full figure caption, which makes it hard to take much away from it. Moreover, the claim hinges on the assumption that <x_n, x_m> is likely to be positive if classes are near the decision boundary. This doesn't seem to be addressed in Figure 4, and I don't believe it is necessarily the case in high-dimensional spaces, where points can be very different and still close to the decision boundary. Are the authors assuming that _all_ inner products of the representations are positive, since they use ReLU activations? This should be explicitly clarified.

In Theorem 2, the authors mention that a deep net can be viewed as a representation extractor, and that ReLU activation would lead to positive inner products for the representations.
1. First, this seems vacuous, since all inner products (even across classes) would be positive.
2. Second, the inner product of the gradients _of the final linear layer_ are the ones that are guaranteed to have positive inner products, but this claim says nothing about the gradients of remaining layers in the network. ~~Since recent work [1] has shown that forgetting primarily occurs at the shallow layers of the network, this does not seem to be a helpful theorem.~~ It is still useful, since recent work [1] showed that forgetting primarily occurs at the deepest layers of the network, but this should be stated clearly.

Then, the authors move to a simple experiment using deep nets, which is supposed to help extend the findings of Theorems 1 and 2 to the deep learning case. However, the connections between these two appear to be very weak and poorly described.
- It is unclear why representations are obtained by concatenating the outputs of all layers, or how this relates to Theorems 1 and 2. Why not use directly the input feature representation, or better yet use only the final hidden representation before the output layer as hinted at in Theorem 2? This choice is not justified.
- Are the gradients also obtained by concatenating all layers' gradients? This _should_ certainly be the case, since we want to learn about the gradient of the entire model.
- It seems that in Theorem 1 we care only about the sign of the gradient: positive representation similarity likely leads to negative gradient similarity. In Figure 2, _all_ points have positive representation similarity, so it's unclear how we can relate these two. It seems that we should now look for decreasing gradient similarity as representation similarity increases between different classes, but this is not stated or analyzed. Also, there is no mention of \beta, \delta, or \alpha, all of which could be explicitly computed and used to analyze these results.

Overall, the Theorems seem to give results only about signs and not distances, whereas the deep learning experiments try to extend the analysis with distances. This means that distances are actually far more useful for relating the findings to metric learning. Since the empirical findings are actually different from the theoretical ones, these experiments should be far more extensive to justify the proposed method, including analyses on more complex data sets, and a much clearer description of exactly how the experiments are carried out.

The take-away from Section 2.1 seems to be that similar representations across different classes lead to conflicting gradients. This is only hinted at in various ways, but I don't believe it is directly stated. Clarifying this would tremendously help lead into Section 2.2. Moreover, it is never explained what the intuition behind this is, which would also be very helpful to increase the reader's ability to follow the paper. I would expect that the intuition is that having similar representations across different tasks would lead the network to somehow confuse the different classes, but I don't believe this is explained.

In Section 2.2, I find the motivation for learning a diverse set of representations for each class somewhat unconvincing. The experiments of Figure 1 show that the learner suffers forgetting on the previous tasks because it only keeps points with similar representation. However, the only reason why this is a problem is that there exist other points with very dissimilar representations, which will then be ignored by the updated margin. If we learned a representation where all points within a class are similar, then that would not seem to be a problem. While this is somewhat analyzed in the experiments in Table 4 via the rho-spectrum, this analysis is quite incomplete. Moreover, Table 4 shows no correlation between forgetting and rho-spectrum, which is what Figure 1 suggests we should expect. This is not discussed in any way.

To add to the confusion in the analysis, the experiments in Section 5 use inconsistent definitions of what the representation is, with all layers being used as the representation in some cases (just like in the analysis of Figure 2), but only the final representation and the logits being used in other cases (which, as an aside, violates the claim that the inner products are necessarily positive).

The empirical results somewhat inconsistently show that the proposed method is better than baselines. However, considering that MNIST-based data sets are usually not very indicative of performance in more complex data sets, I would focus my analysis on the CIFAR-10 and CIFAR-100 experiments. Here, the proposed method was best in one case but not in another. This would require a much more in-depth analysis, but the whole results analysis is limited to a single paragraph.


[1] Ramasesh, V. V., Dyer, E., & Raghu, M. (2020). Anatomy of catastrophic forgetting: Hidden representations and task semantics. arXiv preprint arXiv:2007.07400.

############## Additional feedback ##############

The following points are provided as feedback to hopefully help better shape the submitted manuscript, but did not impact my recommendation in a major way.

Abstract
- DRL is efficient w.r.t. what?
    - The intro seems to suggest it's computational efficiency
    - The first time we actually get to see what this means is at the end of the paper, buried in the last paragraph of Section 5. Since this seems to be such a relevant point (enough to point to it in the Appendix and Introduction), it would be relevant to bring it up earlier and give more details about how it compares to other methods.

Intro
- I wouldn't consider that methods for growing the model size fall in the same category as regularization-based methods. Their focus isn't preserving knowledge of past tasks' models, but instead adding the current task's knowledge into a separate model.
- Paragraph 4: "direction that closest" --> "direction that [is] closest"
- Why such a detailed summary of gradient-based methods? Push to separate Related Work section.
- Again it's unclear what the metric for efficiency is. Context seems to suggest it's computation time, but one could also think it's data efficiency or memory capacity (since the former is commonly important in lifelong learning and the latter is mentioned in this section).

Sec 2.1
- Why two different notations for x_n with and without boldface? They both seem to denote a single data point.
- Theorem 1 is the first time the authors mention negative inner product of the gradients. Before, only the terms "similar" and "diverse" were used. I suggest making this clearer in the explanation throughout.
- Deep learning experiments
    - Is the correlation coefficient between the x axis (representation similarity) and the y axis (gradient similarity)?
    - It would be helpful to clarify explicitly in the text that in Figure 2 the blue dots should be analyzed under Theorem 1 and the orange dots under Theorem 2.
    - The final statement about discriminative representations of task 1 vs task 2 doesn't seem to add much value, so my suggestion would be to drop it or clarify how it's useful to the argument of the paper.

Sec 2.2
- The first sentence in second paragraph seems disconnected from what follows. What's the point of it? Where will the unused information be leveraged?

Sec 3
- Is the representation h_i the same as in the experiments of Figure 2, i.e., the concatenated output of all layers?
    - It is made clear later that the definition of h_i is not consistent throughout the experiments. Here, the authors should clarify this and explain how this definition should be chosen, and why it is not possible to use a consistent definition.
- Figure 3 is missing a y-axis label.
- The effects on the similarities within classes seem to be very small in Figure 3. Is this because \alpha is small?

Sec 4
- The statement that in the incremental class setting the method can work without task boundaries seems vacuous: whenever the learner sees a new class, evidently it corresponds to a new task.
- This is the first time that task-incremental vs class-incremental settings are introduced, without explaining exactly what they are. The manuscript should be self-contained and so it should give a clear description of these settings if they are relevant to the submission (as they clearly are based on the statements in this section).

Sec 5
- Fashin-MNIST --> Fashion-MNIST
- Details of the experimental setting are quite comprehensive (+).
- In addition, We --> In addition, we
- What is the reason the proposed method achieves a better trade-off between forgetting and intransigence? This is not mentioned at all at any point in the paper up to this point.
- Why only compare computation time on MNIST and not on remaining benchmarks? Since computational complexity depends on the representation size, it is thus clearly dependent on the choice of representation, which inconsistently varies across benchmarks. Also, what is the theoretical complexity of the baselines? Are there settings where they would, at least theoretically, be faster? This comparison would strengthen the claim that the method is more efficient.
- I would also like to see an ablative study removing each of the terms in the regularization function in turn. If ER is not implemented using the same sampling strategy as BER, then I'd also like to see the effect of BER training on its own. This would help tease apart the contributions of this work better.

Appendices
- How are the hyper-parameters of DRL chosen in Table 6?


----------- Updates during discussion -----------

I have updated my score from a 4 to a 5 based on the authors' response. My justification is below in my comment to the authors.

---

> ### Comment · AnonReviewer5 · 2020-11-15
> **Correction about which layers cause forgetting.**
>
> I realized that I incorrectly stated that [1] showed that the shallow layers are the ones that cause forgetting, when in fact it is the opposite: the deep layers are those that cause forgetting. Please accept my correction, which I have added to the original review.
>
> This point was only one of the many that influenced my rating, so for now I choose to maintain my original score.

---

> ### Author Response · Authors · 2020-11-20
> **Response to R5's comments (1/4)**
>
> We really appreciate R5 for giving such thorough review comments and will clarify most of the issues in the revision. In the following we address R5’s main concerns.
>
> We first clarify two crucial issues that seem to have informed some of these concerns:
> 1. We do NOT suggest storing samples with most diverse gradients into the memory, it is suggested in GSS (Aljundi et al. 2019). The experiments in Fig. 1 verifies this idea and the results show it may NOT be sufficient for learning new decision boundaries in continual learning.
> 2. The Theorems do NOT require the inner product of representations always to be positive. We analyze in which cases the inner product of gradients tends to be negative by the Theorems. Using ReLU-like activations is taken as a common case in practice for the analysis. If checking the Theorems by assuming the inner product of representations is negative, they are still valid, e.g. according to Theorem 2, negative inner product of within-class representations results in negative inner product of gradients, which indicates non-negative similarity of representations is preferred within classes. We provide more empirical study in Sec.2 in the revision for a better understanding of the theorems.
>
> “The biggest reason I lean towards rejecting this paper is that I am unconvinced by the claims made throughout Section 2. First, the section begins by suggesting that instances with similar gradients are those most useful for generalization. This seems to contradict the findings of Aljundi et al., which chooses to keep samples with maximum gradient diversity in the episodic memory. This apparent contradiction should be carefully discussed and analyzed.”
>
> The final sentence in the first paragraph of section 2 is: “This in turn indicates that samples that lead to the most diverse gradients provide the most difficulty of generalization.”  We see the confusion caused by this phrase and will rephrase it as ‘It indicates that samples that lead to the most diverse gradients provide the most information that is useful for generalization, which is consistent with Aljundi et al. (2019). ‘
>
> “Then, the experiment of Figure 1 suggests that: the gradients with the most diverse gradients are those close to the decision boundary, and in order to avoid forgetting previously seen tasks, samples with diverse representations help maintain good decision boundaries as new tasks are learned. The former claim is further studied in the rest of the section, but why does this experiment suggest that we would like to keep samples with diverse gradients? I keep seeing conflicting indications that we should store samples with diverse or similar gradients. On the other hand, the latter claim is not substantiated beyond these toy experiments. “
>
> We appreciate that there is some subtlety in the experiments of Fig.1, so we will attempt to clarify. The first experiment in Fig1.a identifies which samples produce the most diverse gradients that could be important in generalization.  The results show those samples are close to the decision boundary. Then we tested storing those samples into the memory (Fig1.b) which was actually testing GSS (Aljundi et al., 2019) because those samples are chosen by GSS, we compared the results with storing randomly chosen samples (Fig1.c). The performance of GSS is worse than the random samples since only storing samples with most diverse gradients of the current task may not be sufficient for learning new decision boundaries.  Hence, the experiment suggests it is NOT enough to only store samples with the most diverse gradients. We do NOT suggest storing samples with similar gradients.
>
> “It seems to be specific to incremental-class learning, and would probably not hold for other variants of continual learning that don't require the same model to discern between the new class and the previous ones.”
>
> Although the motivation is from class-incremental learning, the insights can be useful for instance-incremental learning as well, such as Permuted-MNIST tasks, and the experimental results in Sec.5 demonstrate this.  For example, the previously learned decision boundary relies on several dimensions that are not discriminative for new instances, so if the memory only contains samples close to the previous boundary, it may be difficult to learn a new boundary that works for both old and new instances.
>
> “In Theorem 1, the authors claim that points close to the decision boundary are likely to yield negative inner product of their gradients. Figure 4 in the Appendix seeks to illustrate this, and could potentially be very helpful towards understanding this work better, but 1) the figure is missing a y-axis label and 2) the caption is confusing and contradictory between the sub-figure captions and the full figure caption...”
>
> Sorry for the confusing figure. We removed figure 4 from the Appendix and added new experiments and figures in Sec.2 in the revision for a better understanding.

---

> ### Author Response · Authors · 2020-11-20
> **Response to R5's comments (2/4)**
>
> “Moreover, the claim hinges on the assumption that <x_n, x_m> is likely to be positive if classes are near the decision boundary. This doesn't seem to be addressed in Figure 4, and I don't believe it is necessarily the case in high-dimensional spaces, where points can be very different and still close to the decision boundary. Are the authors assuming that all inner products of the representations are positive, since they use ReLU activations? This should be explicitly clarified.”
>
> Theorem 1 says that for samples from different classes, the inner product of gradients gets an opposite sign of the inner product of representations with a probability that depends on the predictions p_n and p_m. The Theorem does not require <x_n, x_m> to always be positive. If <x_n, x_m> is negative and the probability of flipping the sign is high, it is the preferred case since the gradients will not conflict with each other.  We included experiments in Sec.2 in the revision to show how likely the <g_n, g_m> has an opposite sign to <x_n, x_m>  in several cases.
>
> “In Theorem 2, the authors mention that a deep net can be viewed as a representation extractor, and that ReLU activation would lead to positive inner products for the representations.
>   * First, this seems vacuous, since all inner products (even across classes) would be positive.
>   * Second, the inner product of the gradients of the final linear layer are the ones that are guaranteed to have positive inner products, but this claim says nothing about the gradients of remaining layers in the network. Since recent work [1] has shown that forgetting primarily occurs at the shallow layers of the network, this does not seem to be a helpful theorem.”
>
> Theorem 2 says for two samples from the same class, the inner product of gradients gets the same sign as the inner product of representations. First, this Theorem is only about samples from the same class and it does not require <x_m, x_n> to be positive.  Second, this theorem does not lead to the inner product of gradients of the final linear layer being guaranteed positive, we add experimental results to demonstrate this in the revision. Theorem 2 indicates the cross-entropy loss does not cause conflict gradients within classes if <x_m, x_n> is positive. It explains why samples with most diverse gradients are not sufficient to preserve within-class information in the experiments of Fig.1.
>
> “It is unclear why representations are obtained by concatenating the outputs of all layers, or how this relates to Theorems 1 and 2. Why not use directly the input feature representation, or better yet use only the final hidden representation before the output layer as hinted at in Theorem 2? This choice is not justified.”  “the experiments in Section 5 use inconsistent definitions of what the representation is, with all layers being used as the representation in some cases (just like in the analysis of Figure 2), but only the final representation and the logits being used in other cases (which, as an aside, violates the claim that the inner products are necessarily positive).”
>
> We concatenate outputs of all layers as we consider they behave like different levels of representation,  and when higher layers (layers closer to the input) generate more discriminative representations it would be easier for lower layers to learn more discriminative representations as well. This method also improves the performance of dense nets.  For reduced ResNet18 we found that including outputs of all hidden layers performs almost the same as only including the final representations, so we just use the final layer for lower computational cost. For both the dense network and the reduced ResNet18, we concatenate the logits (as the output of the last layer) to the representations, because the logits play an important role in Theorem 1, e.g. if p_m and p_n are less similar, the probability of getting conflict gradients would be smaller for inter-class samples.  Please find more details in the revision.
>
> “Are the gradients also obtained by concatenating all layers' gradients?...”
>
> Yes, the gradients are computed for the entire model.
>
> “The take-away from Section 2.1 seems to be that similar representations across different classes lead to conflicting gradients. This is only hinted at in various ways, but I don't believe it is directly stated. ... Moreover, it is never explained what the intuition behind this is, ... I would expect that the intuition is that having similar representations across different tasks would lead the network to somehow confuse the different classes, but I don't believe this is explained.”
>
> It is stated in page 3 before the Theorems, the 1) statement in italic fonts.  It is motivated by the experiments in Fig.1 and the Theorems are provided to analyze it formally. We restate it here: ‘1) more similar representations in different classes result in more diverse gradients.’

---

> ### Author Response · Authors · 2020-11-20
> **Response to R5's comments (3/4)**
>
> “It seems that in Theorem 1 we care only about the sign of the gradient: positive representation similarity likely leads to negative gradient similarity. In Figure 2, all points have positive representation similarity, so it's unclear how we can relate these two. It seems that we should now look for decreasing gradient similarity as representation similarity increases between different classes, but this is not stated or analyzed. Also, there is no mention of \beta, \delta, or \alpha, all of which could be explicitly computed and used to analyze these results.”“Overall, the Theorems seem to give results only about signs and not distances... This means that distances are actually far more useful for relating the findings to metric learning.”
>
>  Theorem 1 only indicates how likely positive inner product of representations leads to negative inner product of gradients.  According to the new empirical study in the revision, such a case is likely to happen when either or both samples close to the decision boundary. When all points have positive representation similarities, we can try two things to reduce negative gradient similarities: 1). push the representation similarity to zero; 2). optimize the logits to decrease the probability of flipping the sign. This indicates that decreasing similarities of inter-class representations might be useful, which a similar approach to those taken in deep metric learning. The sign is important because only negative inner products of gradients cause conflicts between tasks. The distance is useful in the sense of reducing negative gradient similarities.  We provide demonstrations of the effects of \beta, \delta, \alpha in the revision.
>
> “In Section 2.2, I find the motivation for learning a diverse set of representations for each class somewhat unconvincing. The experiments of Figure 1 show that the learner suffers forgetting on the previous tasks because it only keeps points with similar representation. However, the only reason why this is a problem is that there exist other points with very dissimilar representations, which will then be ignored by the updated margin. If we learned a representation where all points within a class are similar, then that would not seem to be a problem. While this is somewhat analyzed in the experiments in Table 4 via the rho-spectrum, this analysis is quite incomplete. Moreover, Table 4 shows no correlation between forgetting and rho-spectrum, which is what Figure 1 suggests we should expect...”
>
> The point is when a model learns very similar representations within a class for a current task, it may lose information on less similar dimensions in the representation space which may be important to future tasks. Such as the y-dimension is not useful for task 1 in Fig.1a, but it is useful for a second task in Fig.1b & c.  Before the new task comes, the model is unable to learn very similar representations over both dimensions for the first two classes in task 1. Fig.1 is a simplified case to show how the memory affects learning new decision boundaries, in this experiment, the new boundaries happen to have more influence on the old classes and hence reducing rho-spectrum helps alleviate forgetting.  It cannot be guaranteed to always be the case as different datasets have different representation spaces.  In general, reducing the rho-spectrum helps with learning new boundaries, hence, it highly correlates with avg. Accuracy over multiple tasks and shows higher correlations with intransigence than with forgetting, since new boundaries usually have more influence on new tasks.  We included the explanation in the revision.
>
> “The empirical results somewhat inconsistently show that the proposed method is better than baselines. However, considering that MNIST-based data sets are usually not very indicative of performance in more complex data sets, I would focus my analysis on the CIFAR-10 and CIFAR-100 experiments. Here, the proposed method was best in one case but not in another...”
>
> We do not expect DRL can outperform all baselines on all datasets as it is a rather simple method and we demonstrate that it is effective in general, on CIFAR-10 it is still the second best.  According to Tab.2 and 3, Multisimilarity got better avg. intransigence and similar avg. forgetting on CIFAR-10 compared with DRL which indicates Multisimilarity learns better representations to generalize on new classes in this case.  Roth et.al. (2020) also suggests Multisimilarity is a very strong baseline in deep metric learning which outperforms the proposed R-Margin on several datasets. And we use the hyperparameters of Multisimilarity recommended in Roth et.al. (2020) which generally perform well on multiple complex  datasets. We add experimental results of TinyImageNet in the revision, in which case DRL is better than Muiltisimilarity. Overall, our experimental results show that connecting to deep metric learning is a promising direction for continual learning.

---

> ### Author Response · Authors · 2020-11-20
> **Response to R5's comments (4/4)**
>
> “Abstract
> DRL is efficient w.r.t. what?  The intro seems to suggest it's computational efficiency. The first time we actually get to see what this means is at the end of the paper, buried in the last paragraph of Section 5... it would be relevant to bring it up earlier and give more details about how it compares to other methods.”
>
> Thank you for pointing this out! DRL is efficient in terms of computational time and RAM cost, we have moved the discussion about its computational complexity into Sec.3 after the introduction of its formulation in the revision.
>
> “Sec. 2.1
> Is the correlation coefficient between the x axis (representation similarity) and the y axis (gradient similarity)?”
>
> Yes, It is.
>
> “Sec 2.2
> The first sentence in second paragraph seems disconnected from what follows. What's the point of it? Where will the unused information be leveraged?”
>
> We changed the first two sentence of the paragraph in the revision as: "However, the usual concepts in DML may not entirely be appropriate for continual learning, as they also aim in learning compact representations within classes. In continual learning, the unused information for the current task might be important for a future task, e.g. in the experiments of Fig. 1 the y-dimension is not useful for task 1 but useful for task 2."
>
> “Sec. 3
> The effects on the similarities within classes seem to be very small in Figure 3. Is this because \alpha is small?”
>
> Because the loss is mostly from negative pairs that are close to the decision boundary, so in the objective the strength of within classes is relatively weaker, and we set \alpha to 1 in this case.
>
> "Sec. 4
> The statement that in the incremental class setting the method can work without task boundaries seems vacuous: whenever the learner sees a new class, evidently it corresponds to a new task.”
>
> It is not necessarily to have a clear boundary between tasks in an online setting, as when a new class is coming, samples of old classes still can come in.  Such cases will not cause any problem to our method, in comparison, some methods rely on task boundaries to decide when to consolidate learned knowledge of a task and when to train task-specific modules for a new task, such as many regularization-based and architecture-based methods.
>
> "Sec. 5
> What is the reason the proposed method achieves a better trade-off between forgetting and intransigence?”
>
> Because DRL facilitates getting a better rho-spectrum that can help learning new decision boundaries, and as rho-spectrum has high correlation to average accuracy DRL also shows better performance in this criterion which in turn gets a better trade-off between forgetting and intransigence.  We added the explanation in Sec.5 in the revision.
>
> “Why only compare computation time on MNIST and not on remaining benchmarks? Since computational complexity depends on the representation size, it is thus clearly dependent on the choice of representation, which inconsistently varies across benchmarks. Also, what is the theoretical complexity of the baselines? Are there settings where they would, at least theoretically, be faster? ...”
>
> Because we ran experiments with the reduced ResNet18 on a GPU server which is shared by multiple users and the assigned resources are not guaranteed to be the same across all experiments, so for a fair comparison, we didn’t include those results in the paper. Basically, it is a similar case as for the dense network, A-GEM and GSS are much slower than other methods because they need to compute the inner product of gradients. For a larger model, they are even slower. The computational complexity of A-GEM is O(B_r W), where W is the number of parameters of the model, B_r is the reference batch size in A-GEM, and the complexity of GSS is O(B B_m W), where B is the batch size, B_m is the memory batch size in GSS. Representation-based methods have the same computational complexity as DRL (O(B^2 H)) where H is the representation size. Please note that W >> H, e.g. for the reduced ResNet18, W = 1094750, whereas H = 160+dim(logits).
>
> “I would also like to see an ablative study removing each of the terms in the regularization function in turn. If ER is not implemented using the same sampling strategy as BER, then I'd also like to see the effect of BER training on its own. This would help tease apart the contributions of this work better.”
>
> We have tested standalone ER and BER for all benchmarks and the results have been provided in Tab. 1-3.  We also ran DRL+ER but found the results to be strictly worse than DRL+BER so we omitted them.  We included the results of comparing DRL with different replay strategies in Appendix F in the revision.
>
> "Appendices
> How are the hyper-parameters of DRL chosen in Table 6?”
>
> We use 10% training data as validation set to choose hyperparameters by grid search. We give the search ranges of hyperparameters in appendix H in the revision.

---

> > ### Comment · AnonReviewer5 · 2020-11-20
> > **Good response. Still not ready for publication, but improved. (1/2)**
> >
> > I thank the authors for their very comprehensive response. Please find my comments on it below.
> >
> > 1. My concern is not that you suggest storing samples with diverse gradients into memory, but that the text somewhat confusingly talks about the benefits of storying samples with similar gradients. For example: "when more of the gradients point in the same direction, the variance will be smaller, leading to a larger GSNR, and consequently, improved test-time performance" and "samples that lead to the most diverse gradients provide the most information that is useful for generalization". I appreciate that there is a lot of subtlety to what exactly should be stored in memory, but precisely for this reason the word choice is very important in this context.
> >
> > 2. My concern is also not that the theorems require positive $\langle x_n, x_m \rangle$, but that the initial analysis of each theorem was only for the positive case, giving no insight about other settings. The new analysis goes a long way towards addressing this issue.
> >
> > - I think the paper overall improved quite a bit with the additional analysis on Theorems 1 and 2. The connection to the experiments of Figure 2 is also improved.
> >   - The new added experiments in Figure 2 and Table 1 are very useful. They now take a much more nuanced look at what changes across gradients in different cases, and this in turn makes interpreting the theorems much easier and clearer.
> >   - The end of Section 2.1 now clarifies that the logits are also used as part of the representation, just like in the experiments of Section 5, to decrease the probability of getting conflicting gradients. But this seems to be unmotivated: isn't the cross-entropy loss already precisely doing that?
> >
> > - However, I still find the motivation for decreasing within-task similarity to be very unconvincing.
> >   - The motivation is from the experiments of Figure 1, where the authors find that keeping more diverse samples within each class helps reduce forgetting, but then mention in their response that this need not be consistent across data sets, and that in the experiments it is correlated with accuracy instead. This toy experiment is quite incomplete compared to the motivation for reducing similarity across classes. Also, it relies on storing points with similar representations _and_ leaving out dissimilar points. If the method learned to map all within-class samples to similar representations, it seems intuitively that this would no longer happen. A proper discussion and empirical study of this is warranted and lacking from the current draft.
> >   - On the other hand, while empirical evidence in Table 5 _seems_ to suggest that diverse samples help improve accuracy, this is never explicitly measured. The authors could have ablated this regularization term, as I suggested in my original review, and shown that it is indeed necessary for higher performance. This could go a little further in justifying the inclusion of the $\mathcal{L}_{wi})$ term in Equation 2. However, note that higher performance is _not_ a proper motivation; motivation should come from insight about _how_ the added term may help the learning process, which again seems incomplete.
> >
> > - The new computational complexity analysis is very valuable, and makes the contribution towards efficiency much clearer. However, unless I'm missing something, this should be the _additive overhead_ of each of the algorithms on top of the cost of backpropagation to compute the gradients, which is $O(WB)$. This makes the differences more subtle. Moreover, the example pointed out by the authors in their response using ResNet18 considers $H$ only on the final layers, but $W$ on all layers. While we would still have $H < W$, the numbers pointed by the authors might be somewhat misleading.
> >
> > - In the updated results, I find the TinyImagenet results somewhat concerning. While the proposed method outperforms the baselines, all the values are so low that the conclusion seems to be that none of the methods work at all in that setting, including the authors' DRL. Once more, there is no analysis in the updated revision. The results analysis is overall a single paragraph, focused only on discussing the effect of the rho-spectrum, but not on actual performance. I again urge the authors to more carefully analyze their results.

---

> > > ### Comment · AnonReviewer5 · 2020-11-20
> > > **Good response. Still not ready for publication, but improved. (2/2)**
> > >
> > > - I find the following discussion quite useful, and encourage the authors to include their justification for using all layers' outputs to their draft.
> > >     - "We concatenate outputs of all layers as we consider they behave like different levels of representation, and when higher layers (layers closer to the input) generate more discriminative representations it would be easier for lower layers to learn more discriminative representations as well. This method also improves the performance of dense nets. For reduced ResNet18 we found that including outputs of all hidden layers performs almost the same as only including the final representations, so we just use the final layer for lower computational cost. For both the dense network and the reduced ResNet18, we concatenate the logits (as the output of the last layer) to the representations, because the logits play an important role in Theorem 1, e.g. if p_m and p_n are less similar, the probability of getting conflict gradients would be smaller for inter-class samples. Please find more details in the revision."
> > >
> > > - I also appreciate the following discussion. I wasn't suggesting that the method should outperform all baselines in all cases, but merely that the authors should carefully analyze when their proposed method is better and why. This is still missing from the revised draft.
> > >     - We do not expect DRL can outperform all baselines on all datasets as it is a rather simple method and we demonstrate that it is effective in general, on CIFAR-10 it is still the second best. According to Tab.2 and 3, Multisimilarity got better avg. intransigence and similar avg. forgetting on CIFAR-10 compared with DRL which indicates Multisimilarity learns better representations to generalize on new classes in this case. Roth et.al. (2020) also suggests Multisimilarity is a very strong baseline in deep metric learning which outperforms the proposed R-Margin on several datasets. And we use the hyperparameters of Multisimilarity recommended in Roth et.al. (2020) which generally perform well on multiple complex datasets. We add experimental results of TinyImageNet in the revision, in which case DRL is better than Muiltisimilarity. Overall, our experimental results show that connecting to deep metric learning is a promising direction for continual learning.
> > >
> > >
> > > Overall, the authors did a good job in the rebuttal and I believe the updated draft merits a higher score. However, my concerns about the motivation for $\mathcal{L}_{wi}$ and others mentioned above still hold, and so I still don't believe the paper is ready for publication. I will raise my score from the original 4 to a 5.

---

> > > > ### Author Response · Authors · 2020-11-23
> > > > **Thanks for pointing out the missed points**
> > > >
> > > > Thanks for pointing this out. We added the related discussions in the experiment section in the revision.

---

> > > ### Author Response · Authors · 2020-11-23
> > > **Response to R5's remaining concerns (1/2)**
> > >
> > > We thank R5’s insightful comments and timely feedback on our response which helps a lot with improving our paper.  We provided experimental results for addressing the remaining concerns and made amendments according to the feedback. We clarify the issues mentioned in the feedback in the following.
> > >
> > > “My concern is not that you suggest storing samples with diverse gradients into memory, but that the text somewhat confusingly talks about the benefits of storying samples with similar gradients. For example: "when more of the gradients point in the same direction, the variance will be smaller, leading to a larger GSNR, and consequently, improved test-time performance" and "samples that lead to the most diverse gradients provide the most information that is useful for generalization". I appreciate that there is a lot of subtlety to what exactly should be stored in memory, but precisely for this reason the word choice is very important in this context.”
> > >
> > > Thanks for further explaining the confusion caused by the wording. We rephrase it in the revision as: “when more of the gradients point in diverse directions, the variance will be larger, leading to a smaller GSNR, which indicates that reducing the diversity of gradients can improve generalization. This finding leads to the conclusion that samples with the most diverse gradients contain the most critical information for generalization, which is consistent with in Aljundi et al. (2019).”
> > >
> > >
> > >
> > > “The end of Section 2.1 now clarifies that the logits are also used as part of the representation, just like in the experiments of Section 5, to decrease the probability of getting conflicting gradients. But this seems to be unmotivated: isn't the cross-entropy loss already precisely doing that?”
> > >
> > > The cross-entropy loss tries to put more probability on the true class of each sample and DRL tries to decrease the similarity between predictions of two samples, which does not necessarily force more probability mass on the true class. It is still motivated by the same experiments in Fig.2, as $\mathbf{x} \sim S_3$ (close to the decision boundary between two classes) gets larger prediction similarity than $\mathbf{x} \sim S_0$ (close to the intersection of three classes) due to the predictions put most probability mass concentrate on both classes of a pair. We included the explanation into the last paragraph of Sec. 2.1 in the revision.
> > >
> > >
> > >
> > > “However, I still find the motivation for decreasing within-task similarity to be very unconvincing.
> > >
> > > The motivation is from the experiments of Figure 1, where the authors find that keeping more diverse samples within each class helps reduce forgetting, but then mention in their response that this need not be consistent across data sets, and that in the experiments it is correlated with accuracy instead. This toy experiment is quite incomplete compared to the motivation for reducing similarity across classes. Also, it relies on storing points with similar representations and leaving out dissimilar points. If the method learned to map all within-class samples to similar representations, it seems intuitively that this would no longer happen. A proper discussion and empirical study of this is warranted and lacking from the current draft.”
> > >
> > > We appreciate the subtlety in this case. The experiments of Fig.1 demonstrate two issues: 1). We should not put too similar samples into the episodic memory and selecting samples by gradient diversity might lead to this issue. In our experiments, the memory is formed by random samples except for GSS. 2). We should not learn too compact representations in a current task which might omit important dimensions in the representation space for a future task, e.g., omitting the y-dimension would not affect task 1 but would affect task 2. In this case, even if we store diverse samples into the memory, the learned representations may be insufficient to generalize on future tasks as the omitted dimensions can only be relearned by using limited samples in the memory. We demonstrate this through experiments with and without L1 regularization in a MLP for the first two tasks of split-MNIST and split-Fashion MNIST, the results are provided in Tab.2, Sec. 2.2 in the new revision.

---

> > > > ### Author Response · Authors · 2020-11-23
> > > > **Response to R5's remaining concerns (2/2)**
> > > >
> > > > “On the other hand, while empirical evidence in Table 5 seems to suggest that diverse samples help improve accuracy, this is never explicitly measured. The authors could have ablated this regularization term, as I suggested in my original review, and shown that it is indeed necessary for higher performance. This could go a little further in justifying the inclusion of the $\mathcal{L}_{wi}$ term in Equation 2. However, note that higher performance is not a proper motivation; motivation should come from insight about how the added term may help the learning process, which again seems incomplete.”
> > > >
> > > > It seems we misunderstood the original comment regarding the regularization terms. We added the ablation study on the two terms in DRL in the Appendix E.  In general, L_bt  gets a better performance in terms of forgetting,  L_wi gets a better performance in terms of intransigence, and a lower  $\rho$-spectrum, and both of them show improvements on BER (without any regularization terms). Overall, combining the two terms obtains a better performance on forgetting than standalone L_bt and retains the advantage on intransigence that brought by L_wi. This indicates that preventing over-compact representations while maximizing margins can improve the learned representations that provide improved generalization over previous and new tasks.
> > > >
> > > >
> > > > “The new computational complexity analysis is very valuable, and makes the contribution towards efficiency much clearer. However, unless I'm missing something, this should be the additive overhead of each of the algorithms on top of the cost of backpropagation to compute the gradients, which is $O(WB)$ This makes the differences more subtle. Moreover, the example pointed out by the authors in their response using ResNet18 considers $H$ only on the final layers, but $W$ on all layers. While we would still have $H < W$, the numbers pointed by the authors might be somewhat misleading.”
> > > >
> > > > Yes, the computational complexity of all methods is additional to the normal backpropagation.  We added the clarification into the revision. We did not put the numbers into the paper as it depends on the applied model. On the dense net, we have W=89610, H=210, it is a smaller gap compared to ResNet18, but we still find that gradient-based methods are more than twice as slow overall. In practice, it is common to use a model that is larger than ResNet18, which would make gradient-based methods even slower.
> > > >
> > > >
> > > >
> > > > “In the updated results, I find the TinyImagenet results somewhat concerning. While the proposed method outperforms the baselines, all the values are so low that the conclusion seems to be that none of the methods work at all in that setting, including the authors' DRL. Once more, there is no analysis in the updated revision. The results analysis is overall a single paragraph, focused only on discussing the effect of the rho-spectrum, but not on actual performance. I again urge the authors to more carefully analyze their results.
> > > >
> > > > TinyImagenet has much lower performance than others because it has more classes (200), a longer task sequence (20 tasks), and a larger feature space (64x64x3).   We added the accuracy of training a single task over all classes by one epoch on all benchmarks into the revision, which can be viewed as an upper bound of the accuracy and it is 17.8% for TinyImageNet.  According to the results the longer task sequence, more classes, and larger feature space all increase the gap between the performance of a single task and continual learning. We included the discussion in the revision as well.  Please note that we didn’t tune the hyperparameters for TinyImageNet or make any pre-processing.

---

> > > > > ### Comment · AnonReviewer5 · 2020-11-23
> > > > > **Good start to ablative tests, but still unconvincing motivation**
> > > > >
> > > > > Thank you for providing additional answers to my questions.
> > > > >
> > > > > - The added ablative tests on the effect of $\mathcal{L}\_{bt}$ and $\mathcal{L}\_{wi}$ are a good, and provide some proof that both proposed losses are useful. I recommend that the authors replicate these experiments on the remaining benchmarks, where results tend to be more nuanced and give more information. I know that it is infeasible to do so in such a short notice, so this would likely need to go through an additional round of reviews.
> > > > > - Thanks for the further clarification about including the logits in the representation. I found it informative.
> > > > > - Unfortunately, my biggest concern continues to be the weak motivation for incorporating $\mathcal{L}\_{wi}$. While now we have some results (the new Table 7 in Appendix E) that suggest that it is good for performance in some cases, I still find the authors' motivation unconvincing. My original point regarding the difference between Figure 1 and the proposed representation learning approach (i.e., that in Figure 1 there exist points with diverse representations in the input space, while the proposed method could learn to map _all_ points to similar representations) still hasn't been addressed by the authors' response. Moreover, I don't see what conclusion we could draw from the experiments in Table 2. There are multiple reasons why adding $\ell\_1$ regularization might lead to worse performance, and using this as evidence that learning compact representations is harmful for continual learning seems to be a stretch. I encourage the authors to continue down this line, attempting to find more convincing explanations about _why_ learning diverse within-class representations might be helpful for continual learning.

---

> > > > > > ### Author Response · Authors · 2020-11-24
> > > > > > **Thanks for R5's feedback**
> > > > > >
> > > > > > Thanks for R5's feedback, we understand the concern about the ablative study.
> > > > > > However, we found it a bit confusion about the concern with the motivation of L_wi  "i.e., that in Figure 1 there exist points with diverse representations in the input space, while the proposed method could learn to map all points to similar representations". In our understanding, leaning compact representations is to map all points to similar representations,  e.g. if the model learns a representation only having one active dimension which is x-dimension in task 1 in the experiments of Fig.1.  Could R5 please give an example to demonstrate that what is the 'similar representations' that the model could learn?
> > > > > > Many Thanks!

---

> > > > > > > ### Comment · AnonReviewer5 · 2020-11-24
> > > > > > > **Response about similar representations**
> > > > > > >
> > > > > > > We indeed mean the same thing by similar representations. The authors encourage their model to learn _dissimilar_ representations within a class, and motivate that in Figure 1 by showing that if the agent only retains _similar_ points in its replay buffer, then it can't learn well given that small buffer. However, this seems to only be true because there are _other_ points, not stored in the replay buffer, that are very _dissimilar_ from the points the agent stored in its buffer, and the agent can't learn good decision boundaries from that limited data. In a representation learning case, the representations are not fixed (like they are in Figure 1), so even if the agent learned a representation that mapped all points within a class to _similar_ representations, that wouldn't seem to be a problem: storing only similar samples in memory would be representative of the entire distribution over representations, because there wouldn't exist other _dissimilar_ points. Hopefully this clarifies my concern about the motivation of $\mathcal{L}\_{wi}$.

---

> > > > > > > > ### Author Response · Authors · 2020-11-24
> > > > > > > > **Thanks a lot for the clarification**
> > > > > > > >
> > > > > > > > Thanks a lot for R5's further clarification! In our understanding, having a representative memory for the entire distribution seems an ideal case, please also refer to [1].  In continual learning, the forward transferring can only happen when a new task is able to reuse representations that are learned from the previous task and it would make the model performs much better on both tasks. So,  learning compact representations might be detrimental to forward transferring.   Also, new decision boundaries need new representations that can only be learned from the memory which could be less representative for the entire distribution.  We hope this makes some sense for preventing compact representations in continual learning.
> > > > > > > >
> > > > > > > > [1] Knoblauch, Jeremias, Hisham Husain, and Tom Diethe. "Optimal Continual Learning has Perfect Memory and is NP-hard." ICML (2020).

---

### Decision · Program_Chairs · 2021-01-07
**Final Decision**

**Decision:**

Reject

**Comment:**

This paper explores the connection between diversity of gradients and discriminativeness of representations. Based on the observations, authors propose Discriminative Representation Loss (DRL).

This paper resulted in a lot of discussions and specifically, R5's detailed comments helped the authors improve their paper. Authors did a good job in making significant improvements to the paper based on the reviews, including better connecting the theory and experiments. However, after much discussion it was felt that experiments and analysis still needed improvement, leading to a decision to reject. The authors are encouraged to use the reviewers' post-discussion updates to further improve and submit to a future venue.